# KRAS interaction with RAF1 RAS-binding domain and cysteine-rich domain provides insights into RAS-mediated RAF activation

Timothy H. Tran[1,4], Albert H. Chan[1,4], Lucy C. Young[2,4], Lakshman Bindu[1], Chris Neale[3], Simon Messing[1], Srisathiyanarayanan Dharmaiah [1], Troy Taylor[1], John-Paul Denson[1], Dominic Esposito [1], Dwight V. Nissley [1], Andrew G. Stephen[1], Frank McCormick [1,2✉] & Dhirendra K. Simanshu [1✉]

The first step of RAF activation involves binding to active RAS, resulting in the recruitment of RAF to the plasma membrane. To understand the molecular details of RAS-RAF interaction, we present crystal structures of wild-type and oncogenic mutants of KRAS complexed with the RAS-binding domain (RBD) and the membrane-interacting cysteine-rich domain (CRD) from the N-terminal regulatory region of RAF1. Our structures reveal that RBD and CRD interact with each other to form one structural entity in which both RBD and CRD interact extensively with KRAS. Mutations at the KRAS-CRD interface result in a significant reduction in RAF1 activation despite only a modest decrease in binding affinity. Combining our structures and published data, we provide a model of RAS-RAF complexation at the membrane, and molecular insights into RAS-RAF interaction during the process of RAS-mediated RAF activation.

[1]NCI RAS Initiative, Cancer Research Technology Program, Frederick National Laboratory for Cancer Research, Leidos Biomedical Research, Inc., Frederick, MD, USA. [2]Helen Diller Family Comprehensive Cancer Center, University of California, San Francisco, CA, USA. [3]Theoretical Biology and Biophysics, Los Alamos National Laboratory, Los Alamos, NM, USA. [4]These authors contributed equally: Timothy H. Tran, Albert H. Chan, Lucy C. Young. ✉email: frank.mccormick@ucsf.edu; dhirendra.simanshu@fnlcr.nih.gov

The RAS-RAF-MEK-ERK signaling pathway regulates diverse cellular processes, including cell proliferation, differentiation, and survival. The interaction of membrane-bound active RAS with RAF is the first step in the activation of this pathway[1,2]. Active RAS recruits RAF to the membrane where RAF dimerizes and becomes active. The activated RAF kinases then phosphorylate and activate MEK kinases, which in turn phosphorylate and activate ERK kinases. Finally, activated ERK kinases are translocated into the nucleus, where they phosphorylate multiple substrates, including transcription factors. RAS mutations resulting in constitutive activation of the MAPK pathway are the most common mutations in human cancers. Strategies that aim to prevent RAS-mediated RAF activation are being developed to treat RAS-driven cancers[3].

The RAF family comprises three evolutionarily conserved cytosolic serine/threonine kinases (ARAF, BRAF, and RAF1, also known as CRAF), which share three conserved regions, CR1, CR2, and CR3[4] (Fig. 1a). The N-terminal CR1 region contains the RAS-binding domain (RBD) and the cysteine-rich domain (CRD), the CR2 region contains a 14-3-3 recognition site, whereas the C-terminal CR3 region includes the kinase domain and a second binding site for 14-3-3 proteins. Prior to RAF

activation, a 14-3-3 dimer binds to phosphorylated serines present in CR2 and CR3, keeping RAF in an autoinhibited state. A recent cryoEM structure of this state revealed that the CRD interacts with both 14-3-3 and the kinase domain of RAF in a manner that precludes CRD interaction with the membrane[5].

Membrane-anchored active RAS binds to the RBD with nanomolar affinity. Following the recruitment of RAF to the membrane, the CRD forms contacts with phospholipids, releasing from the autoinhibited state. Based on mutagenesis and NMR experiments, CRD residues proposed to be involved in membrane interaction are in the two hydrophobic loops surrounded by basic residues[6–8]. The CRD has been reported as a second RAS-binding domain of RAF[9–13], that can bind to RAS independently of RBD with weaker (micromolar) affinity[14]. Interaction between CRD and RAS is critical for activation of RAF[12,15–18]. RAS residues N26 and V45, and CRD residues S177, T182, and M183 play an essential role in RAS-CRD interaction[10,12]. NMR and mutagenesis studies also suggested that CRD may interact with RAS residues 23–30[13], residues in the switch-II region (G60 and Y64)[11], as well as the farnesyl group of RAS[13,14]. The CRD, therefore, plays multiple roles in the RAF activation cycle: it maintains RAF in the autoinhibited state, and is critical in the process of RAF activation

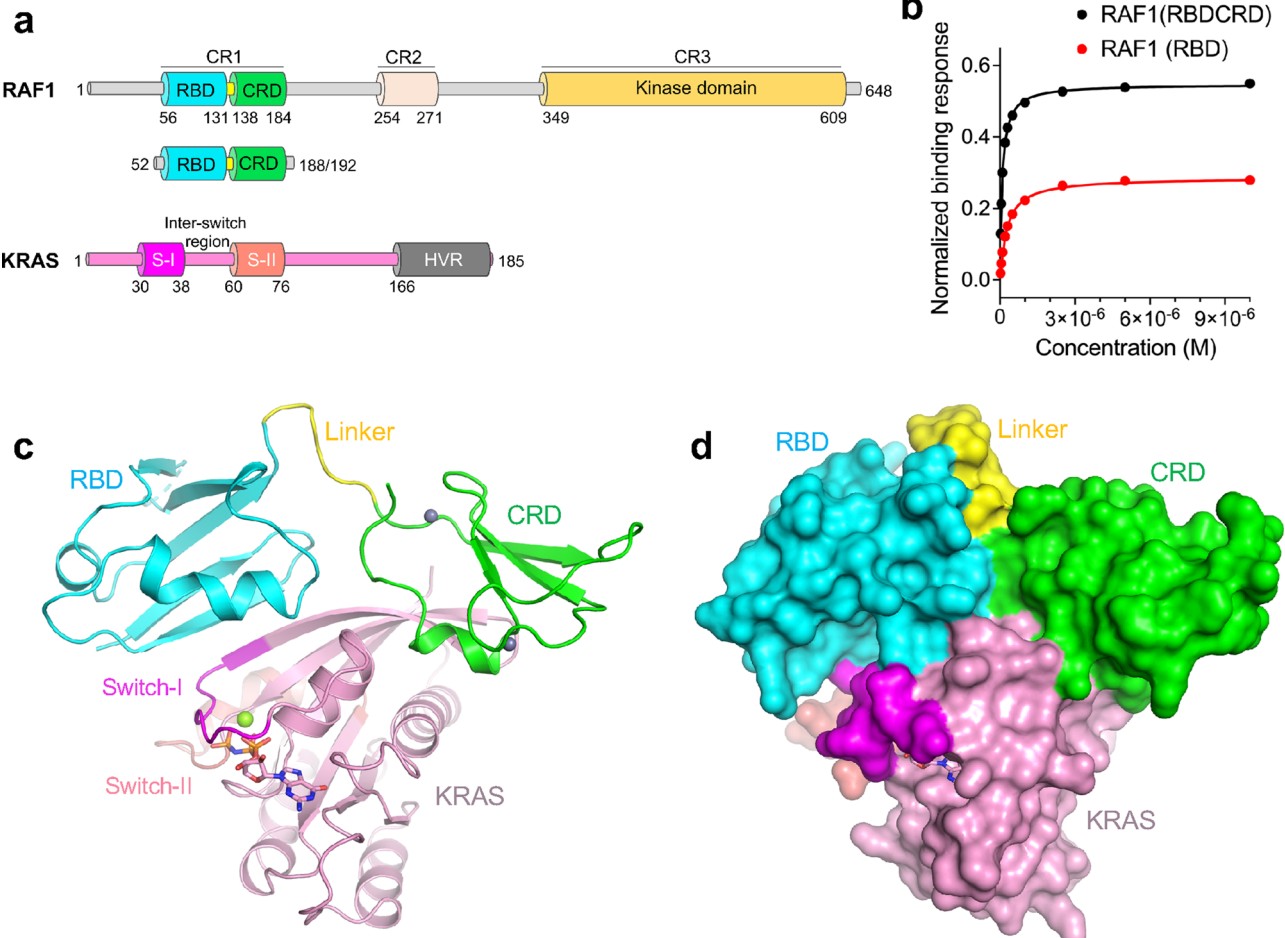

**Fig. 1 Structure of the KRAS-RAF1(RBDCRD) complex and SPR analysis. a** Domain architecture of RAF1 and KRAS. Key domains and their boundaries are shown. The RAF1 construct containing RBD and CRD regions used in this study is shown below the full-length RAF1. The G-domain of KRAS4b(1–169) used in this study contains switch-I (S-I) and switch-II (S-II) regions with the inter-switch region present between them. **b** Steady-state binding isotherms derived from the representative SPR experiments used for measuring the binding affinities of RAF1(RBDCRD) and RAF1(RBD) to KRAS-GMPPNP. Source data are provided as a Source Data file. **c**, **d** The overall structure of the complex formed by GMPPNP-bound KRAS and RAF1(RBDCRD) in **c** cartoon, and **d** surface representations. GMPPNP is shown as sticks, and Mg$^{2+}$ (green) and Zn$^{2+}$ (gray) are shown as spheres. KRAS is colored pink with switch-I and switch-II regions highlighted in magenta and salmon, respectively. RBD, CRD, and the linker present between them are colored cyan, green and yellow, respectively.

through interactions with the plasma membrane and with RAS itself.

Although crystal structures of HRAS and Rap1 in complex with RBD and an NMR structure of an isolated CRD have been solved[19–21], no experimental structure of the RAS-RBDCRD or RAS-CRD complex is currently available. Recent reports of the cryoEM structures of full-length BRAF in complex with 14-3-3 and MEK have provided insights into the autoinhibited and active RAF complexes. However, none of these structures showed the first half of BRAF in its entirety[5,22], presumably because conserved regions CR1 and CR2 are too flexible in the absence of RAS. While progress has been made in understanding the role of RAS in RAF activation, the molecular basis of RAS-RAF binding via the formation of a RAS-RBDCRD complex at the cell membrane, and the role of CRD in stabilizing the active RAS-RAF complex are not fully understood.

Here, we present crystal structures of wild-type and oncogenic mutants of KRAS in complex with the RBD and CRD of RAF1. The structures provide detailed insights into the KRAS-CRD interaction interface and show how RBD and CRD are arranged with respect to one another. They also extend the footprint of RAF binding to RAS beyond switch I, to residues in the inter-switch region and helix α5. The interswitch region differs significantly amongst RAS-related proteins in the RRAS, RIT, and RAP families and may explain why all of these proteins can bind RAF kinases, but only HRAS, KRAS, and NRAS proteins activate RAF kinases efficiently. We use structure-based mutagenesis studies to identify the role of various interfacial residues in KRAS-RBDCRD interaction as well as RAS-mediated RAF activation. Extending published mutagenesis, NMR and molecular dynamic (MD) simulation studies, we propose how the KRAS-RAF1(RBDCRD) complex (hereinafter referred to as KRAS-RBDCRD) might interact with the membrane. Combining the structural insights obtained from our KRAS-RBDCRD structures and the recently solved autoinhibited state of BRAF in complex with 14-3-3 and MEK, we show molecular details of RAS-RAF interaction during the process of RAS-mediated RAF activation.

## Results

### Structures of KRAS in complex with two RAS-binding domains of RAF1.
To understand how the tandem domains, RBD and CRD, in the N-terminal regulatory part of RAF1 interact with KRAS, and their roles in RAS-mediated RAF activation, we attempted to crystallize and solve the structure of KRAS in complex with the RBDCRD region of RAF1 (Fig. 1a). We made multiple constructs of RAF1(RBDCRD). Among these, two constructs with residues ranging from 52–188 to 52–192 yielded soluble and stable proteins after expression and purification. In addition, we expressed and purified RAF1(RBD) (residues 52–131). Measurement of binding affinities by surface plasmon resonance (SPR) showed high-affinity interaction ($K_D \sim 356$ nM) between KRAS and RAF1(RBD), as shown previously[23]. The presence of CRD in the RAF1(RBDCRD) resulted in a more than two-fold increase in the binding affinity with KRAS ($K_D \sim 152$ nM) suggesting that CRD also plays a role in KRAS-RBDCRD interaction (Fig. 1b and Supplementary Fig. 1a).

We crystallized the GTPase domain of KRAS (residues 1–169) in complex with RAF1(RBDCRD) (residues 52–188) and solved the structures in two different crystal forms, I and II, at a resolution of 1.95 Å and 2.50 Å, respectively (Table 1 and Supplementary Table 1). Despite having different crystal packing interactions, the structures of both crystal forms are very similar, with r.m.s. deviation of 0.40 Å for Cα atoms or 0.45 Å for all atoms (Supplementary Fig. 1b). Since crystal form I was solved at a higher resolution, subsequent analyses are based on form I unless stated otherwise.

The overall structure shows that the two tandem domains of the RAF1 CR1 region, RBD and CRD, directly contact one another to form one structural entity, and RBD as well as CRD interact with KRAS to a similar extent (Fig. 1c, d). Similar to what has been observed in the HRAS-RAF1(RBD) and Rap1-RAF1 (RBD) structures[19,21], the KRAS-RBD interaction interface is mainly formed by the switch-I region, where RBD and KRAS interact mainly via β-strands and form an extended β-sheet structure. Interestingly, the CRD interaction with KRAS does not involve the switch regions. Instead, the KRAS-CRD interface is formed by KRAS residues present in the interswitch region and the C-terminal helix α5 (Supplementary Fig. 1c). Like the KRAS-RBD interface, KRAS residues involved at the KRAS-CRD interaction interface are conserved across all four RAS isoforms (Supplementary Fig. 1c). It has been suggested that RBD and CRD form two independent globular domains, where RBD's role is to interact with KRAS, and CRD helps anchor RAF1 to the membrane. Our results suggest that CRD plays not only an important role in anchoring RAF1 to the membrane, but also a direct role in enhancing RAS-RAF interaction.

### KRAS interaction with RBD plays a major role in the high-affinity KRAS-RAF1 complex formation.
To understand what impact the presence of CRD may have on the KRAS-RBD interaction, we solved the structure of KRAS in complex with RAF1(RBD) and compared it with the KRAS-RBDCRD structure (Table 1 and Supplementary Table 1). Structural superposition of KRAS-RAF1(RBD) (hereinafter referred to as KRAS-RBD) with KRAS-RBDCRD using KRAS residues shows no significant differences at the KRAS-RBD interaction interface in the two structures (Fig. 2a). However, RBD residues that are located away from the KRAS-RBD interface show significant conformational changes. Structural analysis shows that the RBD domain in the KRAS-RBDCRD structure undergoes a rotation of 9.3° and a displacement of 0.9 Å compared to that of the KRAS-RBD structure (Fig. 2b). This observation suggests that the KRAS-RBD complex is likely to be flexible in the absence of CRD, and this rotation of RBD may be needed to properly orient CRD to stabilize the KRAS-RBDCRD interaction. This is consistent with an MD simulation study which suggested that CRD reduces the fluctuations of the KRAS-RBD complex at the membrane and enhances its binding affinity[24].

The KRAS-RBD interface is mainly composed of hydrogen bonds and electrostatic interactions, where acidic residues located in and around the switch-I region (residues from I24 to R41) of KRAS interact with basic residues present in the β2 strand (N64-V69) and α1 helix (K84-R89) of RBD (Fig. 2c–e and Supplementary Fig. 2a–c). A total of nine hydrogen bonds and four salt-bridges at the KRAS-RBD interface contribute to the nanomolar affinity between KRAS and RBD (Fig. 2c). Most of these interactions are observed in both KRAS-RBD and KRAS-RBDRCD structures (Supplementary Tables 2 and 3). In addition, crystal form II of the KRAS-RBDRCD structure contains an extra salt-bridge between RBD R59 and KRAS E37. These interactions resemble the protein-protein interactions observed in the previously solved structures of HRAS-RAF1(RBD) and Rap1-RAF1(RBD)[19,21].

To understand the role of RBD residues in the KRAS-RBDCRD interaction interface, we mutated five RBDCRD residues (R59A, N64A, Q66A, R89L, and F130E) and examined their binding affinity to active WT-KRAS (GMPPNP-bound) using an SPR assay (Fig. 2f, g and Supplementary Fig. 2d). Analysis of the binding affinity shows that R59A, N64A, and Q66A mutants in RBDCRD resulted in a 3–12-fold reduction in the binding affinity with KRAS, whereas mutation of R89L resulted in almost complete loss of interaction between RBDCRD

**Table 1 Crystallographic data collection and refinement statistics.**

| | KRAS-RAF1-RBD | KRAS-RAF1-RBDCRD (Crystal form I) | KRAS RAF1-RBDCRD (Crystal form II) | KRAS[G12V]-RAF1-RBDCRD | KRAS[G13D]-RAF1-RBDCRD | KRAS[Q61R]-RAF1-RBDCRD |
|---|---|---|---|---|---|---|
| *Data collection* | | | | | | |
| Resolution range (Å) | 44.14–1.40 (1.45–1.40) | 44.17–1.95 (2.02–1.95) | 48.65–2.50 (2.59–2.50) | 43.24–2.87 (2.97–2.87) | 43.45–2.11 (2.19–2.11) | 48.32–2.70 (2.80–2.70) |
| Space group | P 2$_1$2$_1$2$_1$ | P 6$_3$22 | P 622 | P 622 | P 622 | P 622 |
| *Unit cell* | | | | | | |
| a, b, c (Å) | 44.13, 67.57, 71.26 | 91.89, 91.89, 154.11 | 133.98, 133.98, 89.28 | 132.10, 132.10, 90.17 | 132.75, 132.75, 89.48 | 132.32, 132.32, 89.91 |
| α, β, γ (°) | 90, 90, 90 | 90, 90, 120 | 90, 90, 120 | 90, 90, 120 | 90, 90, 120 | 90, 90, 120 |
| Total reflections | 387,827 (26,517) | 264,233 (42,280) | 102,583 (16,570) | 63,877 (10,610) | 354,688 (56,895) | 161,066 (24,222) |
| Unique reflections | 81,103 (6024) | 28,734 (4490) | 16,642 (2645) | 10,931 (1720) | 27,556 (4174) | 13,291 (2070) |
| Multiplicity | 4.8 (4.4) | 9.2 (9.4) | 6.2 (6.3) | 5.8 (6.2) | 12.9 (13.6) | 12.1 (11.7) |
| Completeness (%) | 99.9 (100.0) | 99.8 (99.3) | 98.4 (99.2) | 98.5 (99.7) | 99.1 (95.9) | 99.7 (99.1) |
| Mean I/sigma (I) | 12.58 (2.18) | 24.2 (2.14) | 14.03 (1.91) | 16.04 (2.53) | 18.69 (2.16) | 23.64 (2.00) |
| Wilson B-factor (Å$^2$) | 14.02 | 42.01 | 49.57 | 82.28 | 42.27 | 75.34 |
| R-merge | 0.064 (0.687) | 0.047 (1.09) | 0.089 (0.860) | 0.076 (0.885) | 0.082 (1.17) | 0.072 (1.23) |
| R-meas | 0.072 (0.782) | 0.050 (1.16) | 0.097 (0.937) | 0.079 (0.965) | 0.087 (1.55) | 0.076 (1.29) |
| CC1/2 | 0.998 (0.772) | 1.00 (0.769) | 0.999 (0.740) | 0.998 (0.827) | 0.999 (0.824) | 0.998 (0.80) |
| *Refinement* | | | | | | |
| Reflections used | 42,703 (4189) | 28,734 (2773) | 16,638 (1634) | 10,930 (1072) | 27,209 (2653) | 13,241(1279) |
| Reflections used for R-free | 2001 (193) | 2000 (193) | 1664 (163) | 1093 (107) | 1991 (193) | 1324 (128) |
| R-work | 0.1645 (0.2180) | 0.2042 (0.2819) | 0.1863 (0.2904) | 0.2167 (0.2815) | 0.1844 (0.2274) | 0.1873 (0.3739) |
| R-free | 0.1949 (0.2359) | 0.2321 (0.2962) | 0.2213 (0.3594) | 0.2691 (0.4010) | 0.2043 (0.2545) | 0.2357 (0.4619) |
| Number of non-H atoms | 2222 | 2524 | 2583 | 2459 | 2566 | 2483 |
| Macromolecules | 2025 | 2391 | 2442 | 2386 | 2394 | 2411 |
| Ligands | 52 | 51 | 54 | 61 | 59 | 48 |
| Solvent | 145 | 82 | 87 | 12 | 113 | 24 |
| Protein residues | 245 | 295 | 300 | 302 | 302 | 302 |
| RMS—bond length (Å) | 0.006 | 0.008 | 0.003 | 0.006 | 0.009 | 0.010 |
| RMS—bond angle (°) | 0.95 | 0.99 | 1.29 | 1.349 | 1.04 | 1.60 |
| Ramachandran statistics favoured (%) | 99.17 | 97.92 | 97.59 | 94.59 | 98.31 | 96.27 |
| Allowed (%) | 0.83 | 2.08 | 2.41 | 5.42 | 1.69 | 3.73 |
| Outliers (%) | 0.00 | 0.00 | 0.00 | 0.00 | 0.00 | 0.00 |
| Average B-factor (Å$^2$) | 21.41 | 60.37 | 62.28 | 102.87 | 53.67 | 94.65 |
| Macromolecules | 20.78 | 60.58 | 62.40 | 103.16 | 53.75 | 94.92 |
| Ligands | 22.31 | 53.57 | 64.62 | 98.26 | 50.56 | 91.15 |
| Solvent | 29.97 | 56.14 | 57.64 | 73.18 | 53.71 | 75.24 |

Statistics for the highest-resolution shell are shown in parentheses.

and KRAS. These results are consistent with the previous observation that polar and charged residues present on RBD play a key role in forming high-affinity interactions between KRAS and RBD[25]. Conversely, point mutation of a hydrophobic amino acid F130 present at the RBD-linker interface showed no impact on KRAS-RBDCRD interaction.

Amino acid sequence alignment shows that eight out of twelve RBD residues involved at the KRAS-RBD interaction interface are conserved across all three RAF isoforms (Fig. 2h). The other four residues also exhibit identical amino acids at these positions in two of the three RAF isoforms, suggesting that all three RAF isoforms interact with RAS proteins in a similar manner. RAF1 (RBD) residues G90 and L91 form interdomain interactions with CRD and are also conserved across all three RAF isoforms.

**A short linker present between RBD and CRD brings the two domains together.** In the KRAS-RBDCRD structure, RBD and CRD are connected by a short linker (residue 132–137), which brings these two domains together with interdomain contacts, resulting in an extended structure (Fig. 3a, upper panel). Mapping

the KRAS interaction interface on the RBDCRD surface shows that both RBD and CRD contribute significantly to RAF1 interaction with KRAS (Fig. 3a, lower panel). Calculation of the solvent-accessible area shows that 658 Å$^2$ of RBD and 555 Å$^2$ of CRD are buried when RBDCRD is in complex with KRAS, suggesting that both RBD and CRD have similarly sized interaction interfaces with KRAS. Sequence alignment of the linker region between RBD and CRD reveals that five of the six amino acid residues are conserved across all three RAF isoforms (Fig. 3b) suggesting similar interactions between RBD and CRD amongst all RAFs.

Interactions between the linker region and RBD involve a hydrogen bond and hydrophobic interactions between L131 of RBD and V134 of the linker region via atoms in the main chain and side chain, respectively (Fig. 3c and Supplementary Table 4). Furthermore, the phenyl ring of F130 (RBD) is present in a hydrophobic pocket formed by two linker residues P135 and L136. The linker and CRD interactions involve hydrogen bond and hydrophobic interactions by linker residue T137 and CRD residues T138, and C184 (part of the zinc-finger motif) present at the C-terminal end of CRD. In crystal form II, CRD interaction with the linker region also includes a hydrogen bond between

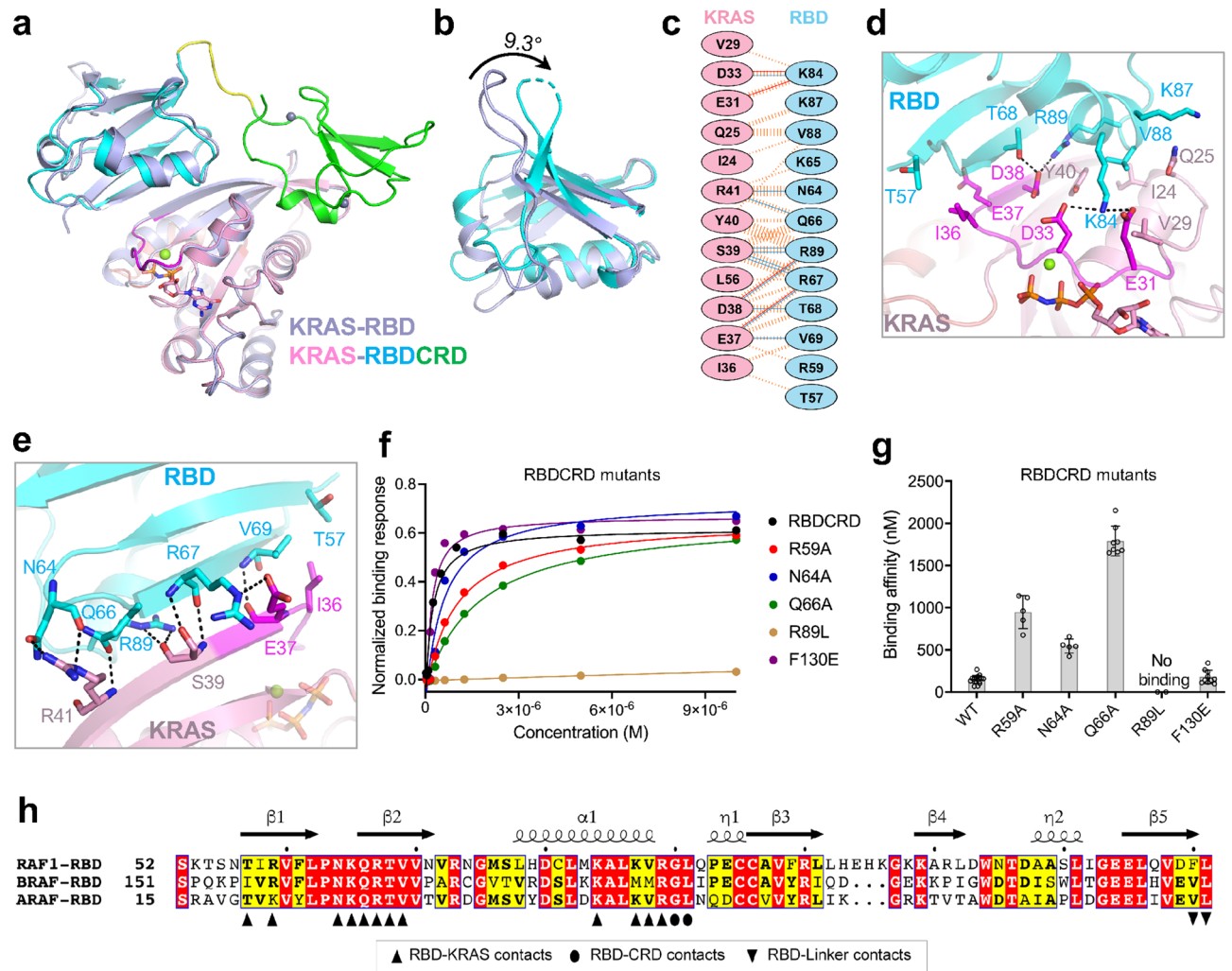

**Fig. 2 Structural and mutational analyses of the KRAS-RBD interactions in the KRAS-RBDCRD complex. a** Superposition of KRAS-RBD structure (light blue) with KRAS-RBDCRD structure shows conformational differences in RBD. The structural superposition was carried out by aligning atoms in KRAS in both structures. **b** Enlarged view of RBD showing conformational differences that occur when it binds to KRAS as RBD vs. RBDCRD. **c** Schematic representation of the KRAS-RBD interaction interface, as identified by PDBSum (http://www.ebi.ac.uk/pdbsum/). The interactions are colored using the following notations: hydrogen bonds—solid blue lines, salt bridges—solid red lines, non-bonded contacts—striped orange lines (width of the striped line is proportional to the number of atomic contacts). **d, e** Enlarged view of the KRAS-RBD interaction interface formed by residues present in the **d** switch-I region (magenta), and **e** β2 strand (residues present at the end of the switch-I region) of KRAS. The nucleotide GMPPNP and residues that participate in the protein–protein interaction are shown in stick representation. Dashed black lines indicate intermolecular hydrogen bonds and salt bridges. **f** Steady-state binding isotherms derived from the SPR data for different point mutants of RBDCRD binding to KRAS-GMPPNP. **g** A bar graph showing binding affinity ($K_D$) obtained using the SPR data shown in **f** for point mutants of RBDCRD located at the KRAS-RBD interface. The $K_D$ values are reported as the mean ± standard deviation from multiple replicates; WT ($n = 17$), R59A ($n = 5$), N64A ($n = 5$), Q66A ($n = 8$), R89L ($n = 4$), and F130E ($n = 10$). Source data are provided as a Source Data file. **h** Amino acid sequence alignment of residues present in the RAS-binding domain (RBD) of human RAF1, BRAF, and ARAF. Fully and partially conserved residues among the RAF isoforms are highlighted in red and yellow, respectively. The secondary structure of RAF1 (RBD) is shown above the alignment. The RBD residues that are involved in the interaction with KRAS, CRD, and the linker region are indicated below the alignment with upright triangles, ovals and inverted triangles, respectively.

residues P135 and D186. The structural superposition of the two crystal forms of the KRAS-RBDCRD complex shows conformational differences in the linker region (Supplementary Fig. 1b), partly due to different crystal packing interactions. It nonetheless shows that the short linker region plays a role in stabilizing RBD and CRD together as one extended structure. RBD and CRD also interact directly with each other via non-bonded interactions formed by residues G90 and L91 from RBD and M183 from CRD (Supplementary Table 4).

To examine the role of hydrophobic linker region residues in stabilizing the RBDCRD structure, we mutated residue L136 within the linker region to alanine. SPR analysis shows that this

L136A mutation led to a 4-fold reduction in binding affinity to KRAS compared with WT-RBDCRD (Fig. 3d and Supplementary Fig. 2d). Considering that L136 is not present at the KRAS-RBDCRD interface, the 4-fold decreased binding affinity for this RBDCRD mutant with KRAS highlights the role of the linker residues in stabilizing RBDCRD and in facilitating interdomain interaction between RBD and CRD.

**RAF1(CRD) interacts with KRAS primarily via the inter-switch region and C-terminal helix α5.** The overall structure of CRD within the tandem RBDCRD present in the crystal structures

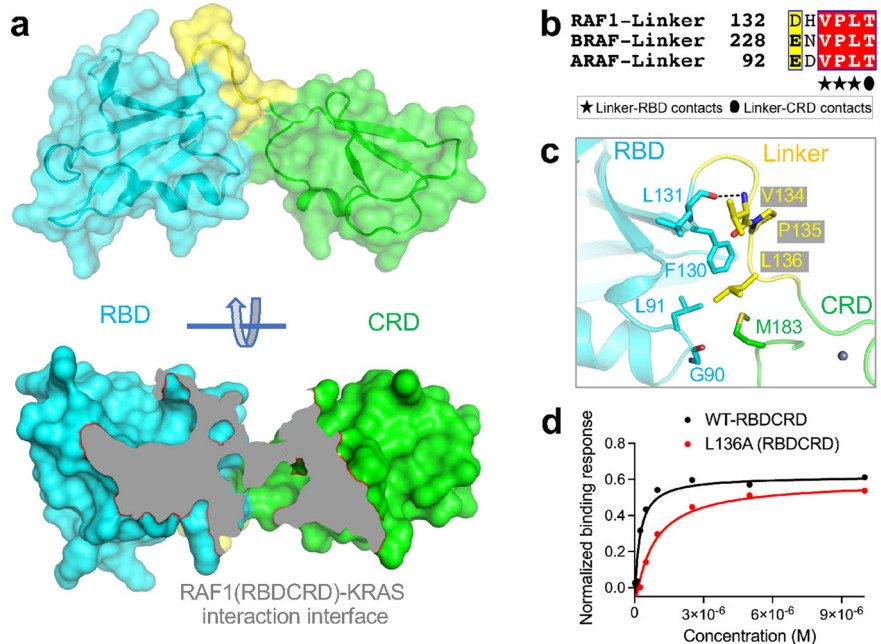

**Fig. 3 Structural and mutational analyses of the role of linker residues present between RBD and CRD in the KRAS-RBDCRD structure. a** The surface representation of RBDCRD showing how linker residues connect RBD and CRD, resulting in the formation of an extended structure. The lower panel shows the region of RBDCRD that interacts with KRAS (colored gray), suggesting RBD as well as CRD interact with KRAS. **b** Amino acid sequence alignment of residues present in the linker region of human RAF1, BRAF, and ARAF. Residues that are involved in the interaction with RBD and CRD are indicated below the alignment with stars and oval, respectively. **c** Enlarged view of the RBD and CRD interaction interface formed by residues present in the linker region. Residues that participate in interdomain interaction are shown in stick representation. A lone hydrogen bond is shown as a black dashed line. **d** Steady-state binding isotherms derived from the SPR data for the WT-RBDCRD and L136A mutant present in the linker region binding to KRAS-GMPPNP.

of the KRAS-RBDCRD complex resembles the previously reported NMR structure of the CRD domain[20] (Supplementary Fig. 3a). The r.m.s. deviation between the crystallographic and NMR (minimized averaged, PDB 1FAR) structure is 1.6 Å for the Cα atoms. Secondary structure assignment for the CRD residues using the program DSSP showed three β-strands (β1–β3) forming an anti-parallel β-sheet and a 3₁₀-helix (η1) present in all structures described here. Similar to the solution structure[20], our crystal structure contains two zinc finger motifs—one near the N- and C-terminal ends of CRD formed by residues H139, C165, C168, and C184, and a second one near the β3 strand and 3₁₀-helix formed by residues C152, C155, H173, and C176 (Supplementary Fig. 3b, c).

The structures of the KRAS-RBDCRD complex provide atomic details of the KRAS-CRD interaction interface, which is similar in size to the KRAS-RBD interface and also consists of nine hydrogen bonds (Supplementary Fig 4a–c). However, unlike the KRAS-RBD interface, which predominantly consists of polar and charged interactions, the KRAS-CRD interface contains no salt bridges and instead includes a relatively large hydrophobic interface (Fig. 4a–c). Interestingly, instead of the switch regions which have been shown to be involved in KRAS interactions with effectors, KRAS interacts with CRD mainly via residues present in the interswitch region (R41, K42, Q43, V44, V45, I46, D47, and G48), and in the C-terminal helix α5 (R149, D153, and Y157) (Supplementary Fig. 3d, e). Seven of these eleven KRAS residues form hydrogen bonds with CRD residues (Fig. 4d, e and Supplementary Table 5). Although the switch regions of KRAS are not involved at the KRAS-CRD interface, KRAS residues L23, I24, and N26 present at the N-terminal end of the switch-I form non-bonded interactions at the KRAS-CRD interface. CRD residues that are involved at the KRAS-CRD interface are present at the N-terminal end of CRD (T138, H139, F141, and R143), β2-strand (F163), 3₁₀-helix (E174, H175, S177, T178, and K179), and

towards the C-terminal end of CRD (V180 and T182). Among these CRD residues, H139, R143, E174, S177, T178, and V180 form hydrogen bond interactions with KRAS. The hydrophobic amino acids in both KRAS and CRD interact with each other and enhance KRAS-RBDCRD interaction. The 3₁₀-helix present in CRD is positioned parallel to the β2 strand and at the top of the α5 helix in KRAS, resulting in surface complementarity at the KRAS-CRD interface.

To identify key residues that play a major role at the KRAS-CRD interaction interface, we mutated hydrophobic and polar residues in both RBDCRD and KRAS, and measured their binding affinities using SPR. Among the CRD point mutants (F141A, E174A, S177N, T178A, K179A, T182A, and M183A) that we tested, K179A and F141A showed 3–4-fold reduced affinity for active KRAS suggesting the importance of both hydrophobic and hydrogen bonding interactions at the KRAS-CRD interface (Fig. 4f, g and Supplementary Fig. 5a). All KRAS mutants (K42A, Q43A, V45E, D153A, and Y157A) that we examined exhibited modest effects, with V45E and to a lesser extent D153A mutations resulting in ~2-fold reduction in the binding affinity (Fig. 4h, i and Supplementary Fig. 5b). The relatively modest effect of mutations at the KRAS-CRD interaction interface suggests that, overall, the interactions between KRAS and RBD play a more significant role in the formation of the KRAS-RBDCRD complex.

Previous studies have suggested that the farnesylated HRAS interacts with CRD in the low micromolar range[12,14]. Our efforts to crystallize farnesylated and methylated full-length KRAS (KRAS-FMe) in complex with RAF1(RBDCRD) failed to give any crystallization hits. To investigate the role of the farnesyl group in the interaction between KRAS and RAF1 CRD/RBDCRD, we carried out SPR binding analysis using active fully processed KRAS-FMe and non-processed KRAS (G-domain) on the chip surface and flowed CRD and RBDCRD over them. RBDCRD bound with an equivalent affinity to non-processed

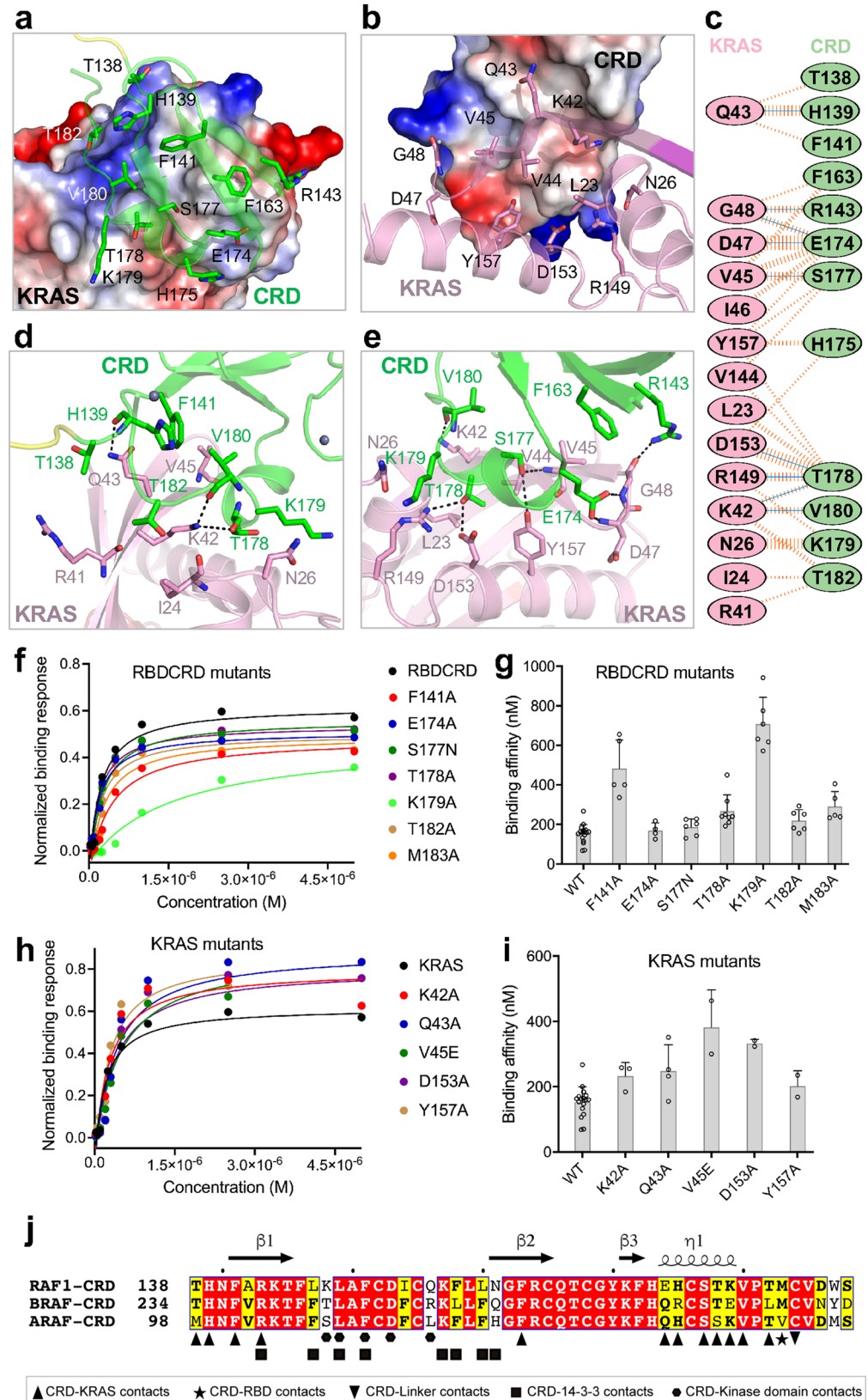

KRAS and fully processed KRAS-FMe ($K_D = 98$ and 128 nM, respectively) whereas CRD showed no binding to non-processed KRAS as well as fully processed KRAS-FMe (Supplementary Fig. 6a, b). In another SPR experiment, we captured CRD and RBDCRD on the chip surface, and flowed active KRAS-FMe or peptide containing farnesylated hypervariable region (HVR-FMe) over the surfaces. KRAS-FMe bound to RBDCRD ($K_D = 84$ nM) but not to CRD, and the HVR-FMe peptide showed no binding to either CRD or RBDCRD (Supplementary Fig. 6c, d). Cumulatively, these experiments indicate that when CRD is present as an isolated domain, we do not detect a binding interaction with either the farnesyl group or G-domain of KRAS under the

**Fig. 4 Structural and mutational analyses of the KRAS-CRD interactions in the KRAS-RBDCRD complex. a** The KRAS-CRD interaction interface, with KRAS shown in electrostatic surface representation and the CRD residues that participate at the interface shown in stick representation. **b** The KRAS-CRD interaction interface, with CRD shown in electrostatic surface representation and the KRAS residues that participate at the interface shown in stick representation. **c** Schematic representation of the KRAS-CRD interaction interface, as identified by PDBSum. The interactions are colored using the following notation: hydrogen bonds—solid blue lines, salt bridge—solid red lines, non-bonded contacts—striped orange lines (width of the striped line is proportional to the number of atomic contacts). **d, e** Enlarged view of the KRAS-CRD interaction interface formed by residues present in the **d** interswitch region, and **e** helix α5 of KRAS. The KRAS and CRD residues that participate in the protein–protein interaction are shown in stick representation. Intermolecular hydrogen bonds are indicated by dashed black lines. **f** Steady-state binding isotherms derived from the SPR binding kinetics for the point mutants of RBDCRD residues (present at the KRAS-CRD interface) binding to KRAS-GMPPNP. **g** A bar graph visualization of binding affinity ($K_D$) obtained using the SPR data shown in **f** for point mutants of RBDCRD residues located at the KRAS-CRD interface. The $K_D$ values are reported as the mean ± standard deviation from multiple replicates; WT ($n = 17$), F141A ($n = 5$), E174A ($n = 4$), S177N ($n = 6$), T178A ($n = 8$), K179A ($n = 6$), T182A ($n = 6$), and M183A ($n = 5$). Source data are provided as a Source Data file. **h** Steady-state binding isotherms derived from the SPR binding kinetics for the mutants of KRAS residues (present at the KRAS-CRD) interface binding to WT-RBDCRD. **i** A bar graph showing binding affinity ($K_D$) obtained using the SPR data shown in **h** for point mutants of KRAS residues located at the KRAS-CRD interface. The $K_D$ values are reported as the mean ± standard deviation from multiple replicates; WT ($n = 17$), K42A ($n = 3$), Q43A ($n = 4$), V45E ($n = 2$), D153A ($n = 2$), and Y157A ($n = 2$). Source data are provided as a Source Data file. **j** Amino acid sequence alignment of residues present in the cysteine-rich domain (CRD) of human RAF1, BRAF, and ARAF. Fully and partially conserved residues among the RAF isoforms are highlighted in red and yellow, respectively. The secondary structure of RAF1(CRD) is shown above the alignment. The RAF1(CRD) residues that are involved in the interaction with KRAS, RBD, and linker are indicated below the alignment with upright triangles, star and inverted triangles, respectively. In addition, based on the cryoEM structure of BRAF-MEK-14-3-3 (PDB 6NYB), CRD residues that are involved in the interaction with 14-3-3 and BRAF kinase domain are indicated with squares and hexagons, respectively.

conditions used in our experiments. However, when CRD is a part of RBDCRD, it increases the binding affinity to KRAS through the formation of multiple hydrogen bonds and non-bonded interactions (Fig. 1b and Supplementary Table 5).

Amino acid sequence alignment of CRD residues in RAF isoforms shows significant sequence identity. Twelve out of fifty residues present in CRD are part of the KRAS-CRD interface (Fig. 4j). Among these twelve residues, six are fully conserved among all three RAF isoforms, whereas the remaining six are conserved among two of the three RAF isoforms suggesting a conserved RAS-CRD interface. In CRD, residue M183 interacts with RBD via hydrophobic interaction, while C184 interacts with T137 located in the linker region via hydrophobic interaction and hydrogen bonding with main chain atoms. Residue C184 is part of the zinc finger and is conserved among all RAF isoforms, whereas M183 is conserved in RAF1 and BRAF and replaced by another hydrophobic residue (valine) in the case of ARAF. A recently solved cryo-EM structure of autoinhibited BRAF in complex with 14-3-3 and MEK (PDB 6NYB) shows CRD occupying a central position in the complex by interacting with a 14-3-3 dimer and the kinase domain of BRAF[5]. Interestingly, except for R143, none of the 14-3-3-interacting and kinase domain-interacting residues from CRD are part of the KRAS-CRD interface (Fig. 4j). This suggests that the KRAS-binding interface is likely to be partially exposed in the autoinhibited state of RAF.

**RAS-CRD interaction is important for RAS-mediated RAF activation.** The KRAS-RBD interface plays a dominant role in the formation of the KRAS-RAF1 complex, but the CRD is essential for RAS-dependent RAF activation[10]. To determine which specific KRAS-CRD interactions are involved in this process, we examined the effects of point mutations in KRAS and RBDCRD on RAF1 kinase activity induced by KRAS in mammalian cells.

To measure the effects of mutations in RAF1, we co-expressed a series of point mutants with constitutively active KRAS Q61L. We purified RAF1 from cells and measured kinase activity associated with each mutant, using MEK1 as a substrate (Supplementary Fig. 7a). Figure 5a shows that point mutations within CRD did not lead to substantial effects on binding to KRAS, consistent with SPR data described above. However, we observed a reduction (25–50%) in kinase activity with CRD or linker mutants, comparable to the effects of RBD mutants (R59A, N64A, Q66A,

and F130E) (Fig. 5b). For comparison, the classic R89L RBD mutant that fails to bind RAS, and the kinase domain mutant (K375M) reduced kinase activity dramatically. Among the RAF1 mutants, T178A mutation located in CRD and L136A mutation located in the linker region led to the highest reduction (~50%) in kinase activity. Both residues potentially play important roles in proper CRD association with KRAS, with T178 side chain forming two hydrogen bonds with KRAS (Fig. 4c, e) and L136 buried at the RBD-CRD interface (Fig. 3c), potentially orienting CRD to interact with KRAS. This suggests that in addition to the high-affinity binding of RAF1 to KRAS provided by RBD, proper CRD association with KRAS is required for robust RAF1 activation.

Similar experiments were performed to examine the role of KRAS residues that interact with CRD in RAF1 activation. KRAS mutations were introduced into the constitutively active Q61L background and then co-expressed with RAF1. These mutants showed no significant changes in KRAS-RAF1 binding (Fig. 5c). With the exception of Y157A, all mutations were less efficient at activating RAF1 kinase (Fig. 5d and Supplementary Fig. 7b). These results suggest that besides the switch-I region in KRAS, residues present in the interswitch region, especially residues K42 to V45, play an important role in RAS-mediated RAF activation, most likely through KRAS-CRD interactions.

Members of RAS subfamily GTPases such as RRAS, TC21, MRAS, Ral, RheB, Rap1, Rit, and Rin share high sequence homology with three RAS isoforms in the switch-I region (Fig. 5e), the main interacting interface with RAF1(RBD). Despite this sequence homology, these RAS subfamily members show significantly reduced levels of RAF activation compared to RAS isoforms[26]. Interestingly, these members of the RAS subfamily have significantly divergent interswitch region sequences (Fig. 5e). We hypothesize that the reduction of RAF activation levels in other RAS family GTPases is due to the different residues in the interswitch region, as this region accounts for the majority of the KRAS-CRD interactions. To test this hypothesis, we swapped residues from the interswitch region in KRAS with that of MRAS, generating the HTE and HTE-NQWAI mutants in different constitutively active onco-genic KRAS backgrounds. While these mutants retain interaction with RAF1 (Fig. 5f), we observed a marked reduction in their ability to activate the ERK pathway, similar to that of constitutively active MRAS-Q71L. Structural alignment of MRAS (PDB: 1X1S) onto KRAS in the KRAS-RBDCRD complex

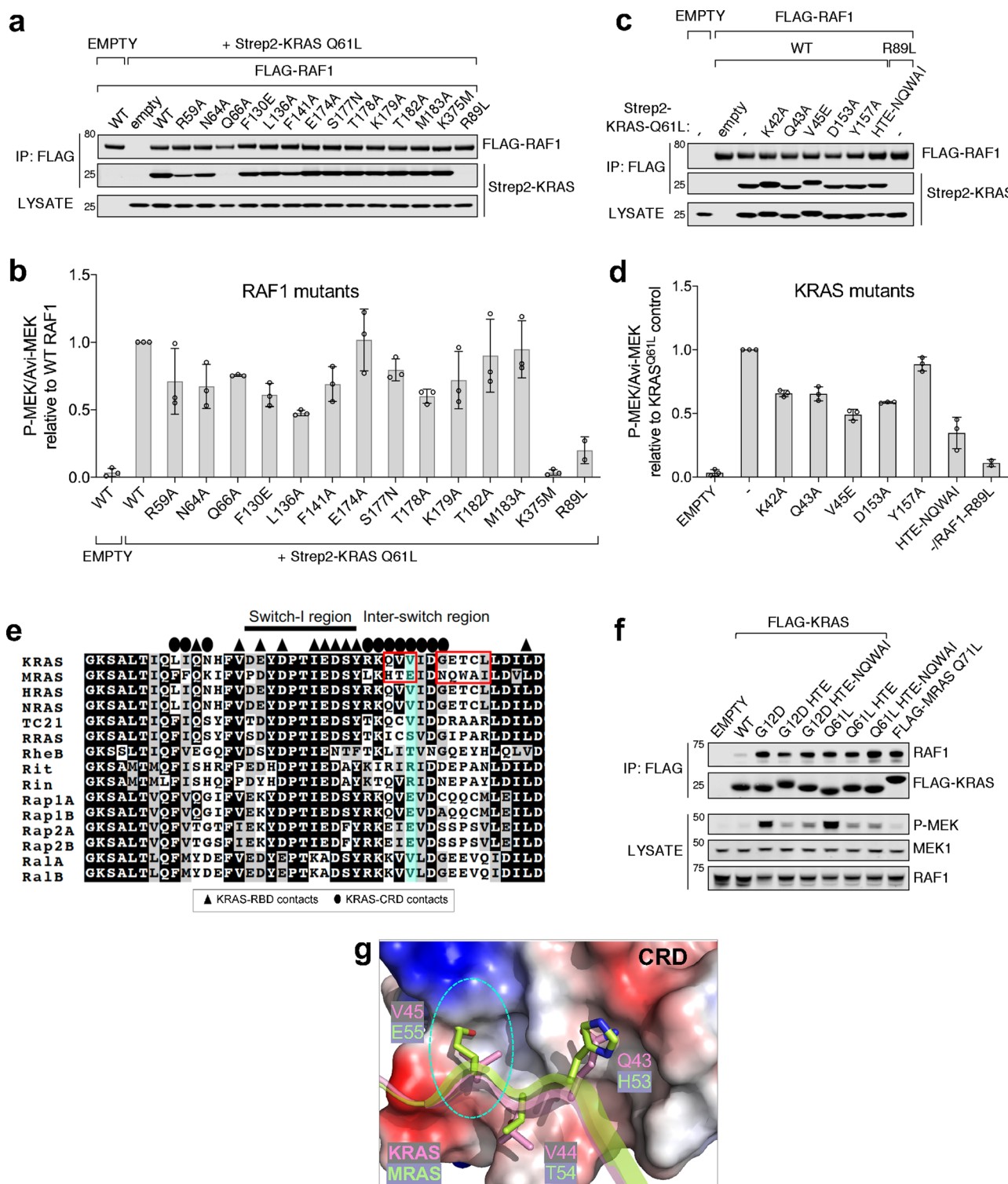

suggests that among the interswitch region mutations, V45E might have the highest impact on RAF1 activation, as the glutamate side chain will likely clash with CRD (Fig. 5g). The importance of V45 is further highlighted by our sequence alignment, which indicates that while V45 is conserved in RAS isoforms, over half of the other RAS subfamily GTPases have long-chain amino acids (glutamate or arginine) instead of valine (Fig. 5e). Taken together, our results suggest that the interswitch region of KRAS is important in fully activating RAF1 by interacting with CRD.

**Oncogenic mutants at G12, G13, and Q61 do not affect KRAS-RAF1(RBDCRD) interaction**. To understand if oncogenic mutations at G12, G13, and Q61 positions in KRAS have any effect on KRAS-RBDCRD interaction interfaces, and to identify any druggable pockets at the KRAS-RBDCRD interface, we attempted to crystallize and solve the structures of G12C/D/V, G13D, and Q61L/R mutants of KRAS in complex with RAF1 (RBDCRD). We were able to obtain well-diffracting crystals of G12V, G13D, and Q61R mutants of KRAS complexed with RAF1 (RBDCRD) and solved their structures in the resolution range

**Fig. 5 Interactions at the RAF1-CRD interface with KRAS are required for full RAF1 kinase activity. a** FLAG-RAF1 WT and mutants for kinase assays were immunoprecipitated with FLAG antibody from 293T cells and assessed for binding to Strep2-KRAS. **b** Kinase activity of purified RAF1 mutants from cells co-expressing KRAS Q61L (shown in panel 5a). **c** FLAG-RAF1 WT and Strep2-KRAS mutants were co-expressed and isolated as in **a**. KRAS mutations were introduced into the constitutively active KRAS-Q61L background and "-" denotes Q61L with no additional mutations. **d** Kinase activity of purified RAF1 co-expressed with KRAS mutants in the constitutively active Q61L background (shown in **c**). **e** Sequence alignment of residues present in the switch I, and interswitch region of human RAS isoforms and members of RAS subfamily. Fully conserved residues are highlighted in black. The switch and inter-switch regions are indicated above the alignment. KRAS residues that are involved at the KRAS-CRD and KRAS-RBD interfaces are highlighted above the alignment using ovals and triangles, respectively. The red boxes and cyan color highlight a lack of conservation in the interswitch region among RAS isoforms and MRAS as well as other members of the RAS subfamily. **f** FLAG-KRAS WT or mutant, or constitutively active MRAS-Q71L immune precipitates were probed for endogenous RAF1 interaction and pathway activation was measured in lysates. "HTE" denotes substitution of inter-switch residues $^{43}$QVV$^{45}$ (KRAS) to HTE (MRAS), and "HTE-NQWAI" was changed from $^{43}$QVV$^{45}$-$^{48}$GETCL$^{52}$ (KRAS) to HTE-NQWAI (MRAS). Representative of $n = 3$ independent experiments. **g** The KRAS-CRD interaction interface, with CRD shown in electrostatic surface representation and the KRAS residues that participate at the interface shown in stick representation. Structural superposition of MRAS (PDB 1X1S) with KRAS shows residue E55 from MRAS clashes with CRD. Bar graphs represent mean phosphorylation ± SD and data points for three independent experiments, except R89L where $n = 2$. Source data are provided as a Source Data file.

between 2.11–2.87 Å. (Table 1 and Supplementary Table 1). Since the mutant structures belong to the same crystal form as the WT KRAS-RBDCRD structure obtained in crystal form II, we used this structure to examine any structural changes in KRAS and at KRAS-RBDCRD interfaces. Structural superposition of WT KRAS-RBDCRD with all three mutant structures shows similar interaction interfaces, suggesting that the two RAS-binding domains of RAF1 bind to oncogenic KRAS mutants similarly to WT KRAS (Fig. 6a). A closer examination of the switch regions of KRAS shows minor conformational changes in the structures of oncogenic mutants complexed with RBDCRD when compared with WT KRAS-RBDCRD structure (Fig. 6b). The presence of a larger side chain at G12, G13, or Q61 position results in local rearrangement of side chains of some of the neighboring residues present in the switch regions without perturbing interactions at KRAS-RBDCRD interfaces. In the KRAS G12V-RBDCRD structure, KRAS residue Y32 adopts a different rotameric conformation to accommodate the side chain of Val at position 12. In the KRAS G13D-RBDCRD structure, the side chain of Y32 is disordered, suggesting multiple rotameric conformations of this residue to avoid a steric clash with the side chain of D13. The side chain of R61 in the KRAS Q61R-RBDCRD structure points towards the KRAS-RBD interface and occupies an empty space surrounded by KRAS residues Y64, P34, and I36, therefore not impacting the KRAS-RBD interface. These results suggest that RAF1 does not differentiate between WT and oncogenic mutants of active KRAS.

**Model showing the KRAS-RAF1(RBDCRD) complex at the membrane.** RAF1(CRD) interacts selectively with phosphatidylserine (PS), and a cluster of basic amino acids, R143, K144, and K148 are critical for this interaction[7,27]. In addition, NMR titration and paramagnetic relaxation enhancement experiments involving RAF1(CRD) binding to nanodisc showed chemical shift perturbations and PRE-induced peak broadening for basic residues K148 and K157, and hydrophobic residues around them[6,8]. Mapping these CRD residues in the KRAS-RBDCRD structure shows that these residues are fully-exposed on the CRD surface, allowing them to interact with the membrane while the CRD is bound to KRAS (Fig. 7a). The electrostatic surface representation of this region of CRD highlights a hydrophobic and basic patch on the surface that is likely to be involved in the interaction with PS-containing membranes (Fig. 7b).

To assess possible membrane interactions of our KRAS-RBDCRD structure, we constructed several membrane-bound active KRAS-RBDCRD models. Previously, MD simulations have identified three highly populated orientations of KRAS on a membrane containing 30% anionic PS lipids in which the region of

RAS that approaches the cell membrane is described by a rotation angle[28] (Supplementary Fig. 8a–c). We aligned our crystal structure of KRAS-RBDCRD to these three KRAS poses and evaluated them based on agreement with CRD residues that have been shown to interact with the membrane[8]. Specifically, the relative arrangement of protein domains in our crystal structure indicates that membrane interaction of KRAS helices α4 and α5 (rotation angle = −15°) is associated with membrane insertion of two CRD loops composed primarily of hydrophobic and cationic residues (Supplementary Fig. 8d, e), in agreement with NMR data[6,8] (Fig. 7c). In contrast, while membrane approach of KRAS β-strands β1–β3 (rotation angle = 105°) allows concerted membrane interaction of KRAS, RBD, and CRD, these orientations exist on the brink of membrane occlusion (Supplementary Fig. 8f) involving RBD and linker regions of RAF (Supplementary Fig. 8g), and are incompatible with membrane insertion of the CRD's hydrophobic loops in the absence of inter-domain reorganization (Supplementary Fig. 8g). Furthermore, KRAS orientation such that helices α3 and α4 approach the membrane (rotation angle = −85°) appears incompatible with existing experimental data since this orientation of KRAS places CRD far from the membrane (Supplementary Fig. 8h). We note that our model of membrane-bound KRAS-RBDCRD shown in Fig. 7c is similar to the State A configuration proposed recently by Fang et al.[6] based on an NMR data-driven model generated using docking software HADDOCK, excepting a 180° rotation of CRD about the approximate RBDCRD long axis (Supplementary Fig. 8i). Whereas the previous model placed CRD helix $^{174}$EHCSTKV$^{180}$ away from KRAS, our crystal structures indicate that this helix makes direct contact with the KRAS G-domain (Supplementary Fig. 9a, b). This discrepancy of the CRD orientation could be due to insufficient NMR distance restraints on the CRD's β1 and 3$_{10}$-helix (since many residues in these regions were broadened beyond detection), the presence of the nanodisc in the NMR experiment, and/or perhaps higher flexibility of CRD in solution in general as indicated by the many broadened peaks. Importantly, membrane interactions of CRD loop residues $^{143}$RKTFLKLAF$^{151}$ and $^{157}$KFLLNGFR$^{164}$ are relatively unaffected when KRAS abuts the membrane with G domain helices α4 and α5 (Supplementary Fig. 9c, d). The membrane-bound KRAS-RBDCRD model shown in Fig. 7c represents a combination of our high-resolution crystal structure with a preferred membrane orientation of KRAS according to MD simulations and NMR data. This model also positions previously identified CRD membrane-interacting residues into the lipid bilayer and may therefore provide a snapshot of RAF1 activation by KRAS at the membrane. However, because this model places KRAS helices α4 and α5 in membrane contact, it is not simultaneously compatible with RAS dimerization at this interface as proposed by Fang et al.[6] in their State B KRAS-RBDCRD model

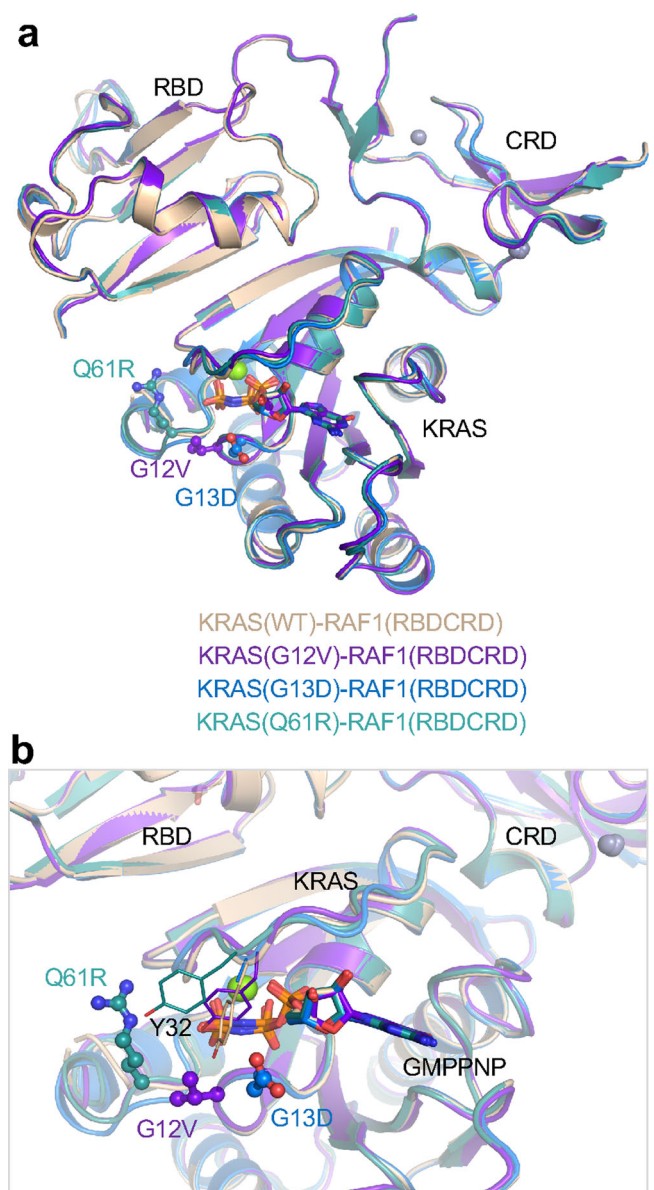

**Fig. 6 Structural comparison of WT KRAS-RBDCRD with the oncogenic mutants G12V, G13D, and Q61R of KRAS bound to RBDCRD. a** Structural superposition of WT, G12V, G13D, and Q61R mutants of KRAS complexed with RBDCRD. The structural superposition was carried out using KRAS residues. Crystal form II of WT KRAS-RBDCRD is used here since it has the same space group as the oncogenic mutants. Oncogenic mutants and GMPPNP are shown in ball-and-stick and stick representations, respectively. The color scheme for different structures is shown in the panel. **b** Enlarged view showing oncogenic mutants, residue Y32 in the switch-I region, and KRAS-RBDCRD interface to highlight similarities and differences among structures of WT and oncogenic mutants of KRAS in complex with RBDCRD.

and others[29,30]. Furthermore, the KRAS-RBDCRD complex may undergo further reorganization driven by membrane association[31].

## Discussion

The RAS-RAF interaction has taken center stage in cancer research as it is implicated in almost 20% of all human cancers[32]. Direct protein-protein interaction between RAS and the N-terminal regulatory domain of RAF is critical for the recruitment

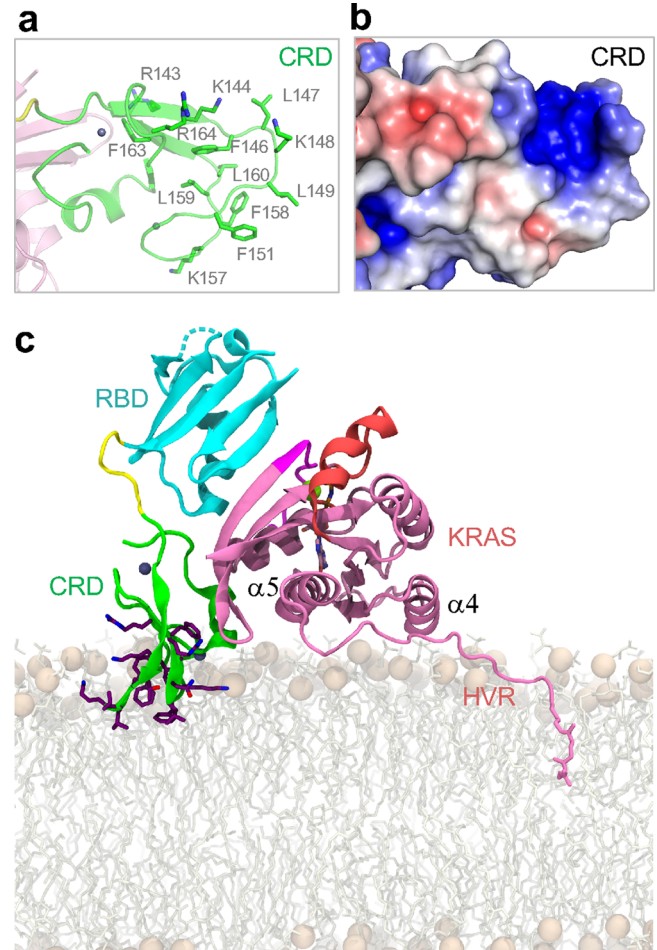

**Fig. 7 Membrane interacting residues in CRD, and a model showing the crystal structure of KRAS-RBDCRD at the membrane. a** Enlarged view of the CRD part of the KRAS-RBDCRD structure with CRD residues (L147, K148, L149, A150, and F158) that have been shown to interact with the PS-containing membrane shown as sticks. **b** Electrostatic surface representation of the CRD shown in **a** highlighting a basic patch which has been proposed to interact with the PS headgroups on the membrane. **c** Crystallographic KRAS-RBDCRD oriented with KRAS helices α4 and α5 in membrane contact, as observed in MD simulations of KRAS[28]. Lipids are sticks with spheres for headgroup phosphorus atoms; KRAS-RBDCRD has the same color-coding defined in Fig. 1c. Side chains of CRD residues [143]RKTFLKLAF[151] and [157]KFLLNGFR[164] are sticks with purple carbon atoms.

of RAF to the plasma membrane, a crucial step in the process of RAS-mediated RAF activation[1,2]. Even though the RAS-RAF interaction was discovered more than 25 years ago, it remains unclear how RAS interacts with RAF during the multistep process of RAF activation. Recent reports of cryoEM and crystal structures of the full-length BRAF or BRAF kinase domain in complex with 14-3-3 and/or MEK have provided insights into the auto-inhibited and active states of RAF[5,22,33,34]. However, in these structures, the entire N-terminal half of BRAF is either not observed or, in the case of the autoinhibited complex, only the CRD can be seen centrally anchored between 14-3-3 and the kinase domain of BRAF[5]. Moreover, these structures do not contain RAS and therefore provide limited insights into RAS-RAF interaction during the process of RAS-mediated RAF activation.

The crystal structure of KRAS in complex with RBDCRD of RAF1 described here shows how the structure of tandem domains RBD and CRD form one structural entity, and how this entity interacts with KRAS. This structure reveals that the KRAS-RBD binding interface is similar to that of HRAS-RBD and Rap1-RBD structures[19,21], as well as KRAS-RBD in the absence of CRD. Importantly, our KRAS-RBDCRD structure provides atomic details of the KRAS-CRD interaction interface, and supports previous mutagenesis experiments suggesting that KRAS residues N26 and V45 contribute to RAS-RAF interaction and RAF activation[12]. Previous mutagenesis studies on RAF1(CRD) have also shown the complete loss of RAS-mediated RAF activation in the CRD double mutants K144A and L160A[10]. These two CRD residues are located in the hydrophobic loops, where they play an important role in anchoring CRD to the membrane. Using the KRAS-RBDCRD structure and MD simulations of KRAS on a membrane with 30% PS[28], we constructed a membrane-bound KRAS-RBDCRD model that has K144, L160 and other residues in the CRD hydrophobic loops inserted into the membrane.

The crystal structure of the KRAS-RBDCRD complex and structural information obtained by comparing it with the auto-inhibited structure of BRAF in complex with 14-3-3/MEK provide insights into the process of RAS-mediated RAF activation (Fig. 8a–c). In the autoinhibited state, BRAF and MEK have been shown to exist as a preassembled quiescent complex bound to 14-3-3[35]. Recent cryoEM structures by Park et al. showed that the BRAF-MEK1 complex is secured in a cradle formed by the 14-3-3 dimer, which binds the two phosphorylated serine sites flanking the BRAF kinase domain[5]. The CRD of BRAF occupies a central position where its membrane-binding surface is buried by interactions with the 14-3-3 dimer and the kinase domain. The molecular handcuffing of BRAF by 14-3-3 therefore maintains the autoinhibited state by blocking kinase domain dimerization and preventing CRD from binding to the membrane. Interestingly, structural superposition of our KRAS-RBDCRD structure to the autoinhibited complex via CRD residues places RBD near the residual electron density observed in the autoinhibited complex, which Park et al. suspect represents RBD[5] (Fig. 8a). In this superposed structure, KRAS can bind to the exposed RBD without sterically clashing with neighboring 14-3-3 or kinase domain residues (Fig. 8b).

Almost all of the CRD residues that interact with KRAS are also exposed in the autoinhibited state and could interact with KRAS. However, the loop containing the phosphorylated serine (S621) at the C-terminus of the kinase domain partially clashes with the KRAS-CRD interaction. In our model of RAF activation by RAS, the activation process starts with the interaction between active KRAS (GTP-bound) and RBD which is exposed in the autoinhibited RAF complex (Fig. 8c). The KRAS-RBD interaction is followed by CRD's release from the autoinhibited complex and its interaction with KRAS and the plasma membrane. It has been suggested that CRD extraction upon RAS binding and its localization to the membrane is the critical event that triggers the release from the molecular handcuffing caused by the 14-3-3 dimer[5]. The release of CRD exposes phosphorylated serine (S259) in the central CR2 region of RAF to be dephosphorylated by the SHOC2-PP1C complex[36], resulting in the formation of an active RAF monomer, where 14-3-3 can no longer bind to CR2 but presumably still binds to the phosphorylated serine in the C-terminal region of RAF.

As shown in the cryoEM structures of the active BRAF complex, 14-3-3 now binds to C-terminal phosphorylated S729 (equivalent to S621 in RAF1) coming from two BRAFs, thus facilitating the formation of an active BRAF dimer. This is supported by recent crystal structures of the BRAF kinase domain in complex with 14-3-3, which showed that dimeric 14-3-3 induces dimerization of the BRAF kinase domain and increases BRAF activity by relieving the negative regulatory effect of ATP which prevents BRAF dimerization[34]. Unlike BRAF, which tends to form homodimers, RAF1 has been shown to prefer to heterodimerize with BRAF[37–39]. Therefore, the formation of the active RAF1 complex is likely to involve heterodimerization between monomeric RAF1 and monomeric BRAF bound to MEK.

Our work, and the work of others, has revealed that CRD plays a crucial role in stabilizing both the autoinhibited and active states of RAF. The importance of this domain is also indicated by the observation that more than 40% of activating BRAF mutations identified in RASopathy syndromes occur in the CRD[40]. Despite being only 50 residues long, CRD shows enormous versatility in that it can interact with the membrane, RAS, 14-3-3, and the RBD and kinase domain of RAF. Discrete interfaces on CRD are arranged in a way that allows this domain to stabilize the autoinhibited state by burying membrane-interacting residues, or conversely facilitate the active state by presenting membrane anchoring residues. Through these interactions, CRD plays a central role in the process of RAS-dependent RAF activation. This role is distinct from its role in binding to RAS, as suggested previously by mutagenesis and biochemical analysis. Mutations in CRD (S177, T182) that prevent interaction with the interswitch region of RAS were described that retain RAS binding via RBD, but are defective in kinase activation or transforming ability[10]. Here, we extend these early observations to show that multiple mutations in the interswitch region of RAS or in CRD can result in impaired kinase activation, often with minimal effects on binding. While RAS has been described as a binary switch, our characterization of CRD indicates that it also plays a binary role in the activation–inactivation cycle of RAF kinases.

The distinction between binding and activation may explain why some members of the RAS superfamily fail to fully activate RAF kinases, despite being able to bind to RAF through the highly conserved switch-I region. We show that, for example, lack of CRD engagement with the interswitch region of MRAS may account for MRAS' inability to activate RAF efficiently. We speculate that the unique interswitch regions of RAS family members may provide specificity for different effector functions, despite the shared switch-I sequences.

Efforts to disrupt the binding of RAS to RAF(RBD) as a therapeutic strategy have, so far, been unsuccessful. The high-affinity RAS-RBD interface, which consists of anti-parallel β-strands, does not offer any obvious pockets to which a small molecule could bind with high affinity. It is also reasonable to expect nonspecific consequences when targeting the RAS-RBD interaction given that this this interface is conserved amongst members of the RAS superfamily. Conversely, targeting the interaction between CRD and RAS, or CRD and the plasma membrane, may offer new therapeutic opportunities, considering their lower binding affinities, the specificity afforded by the interswitch region, and the more flexible nature of the protein interfaces.

## Methods

### Cloning, expression, and purification of recombinant proteins

*Cloning*. Constructs for protein expression were produced using Gateway recombination-based cloning as described previously[41] using attB-flanked *E. coli* optimized synthetic DNA generated by ATUM, Inc. as starting materials. RAF1 (RBDCRD) clones consisted of RAF1 amino acids 52–188 (or 52–192) with an upstream TEV protease site (ENLYFQ); cleavage of this protein with TEV protease leaves the native serine at amino acid 52 as the N-terminal amino acid of the final protein. RAF1(RBD) clone consisted of RAF1 amino acids 52–131 with an upstream TEV protease site (ENLYFQ); cleavage of this protein with TEV protease leaves the native serine at amino acid 52 as the N-terminal amino acid of the final protein. Wild-type and mutants of KRAS4b(1–169) clones were generated with an upstream tobacco etch virus (TEV) protease site (ENLYFQG), which when cleaved during purification leaves an additional glycine residue at the N-terminus of the

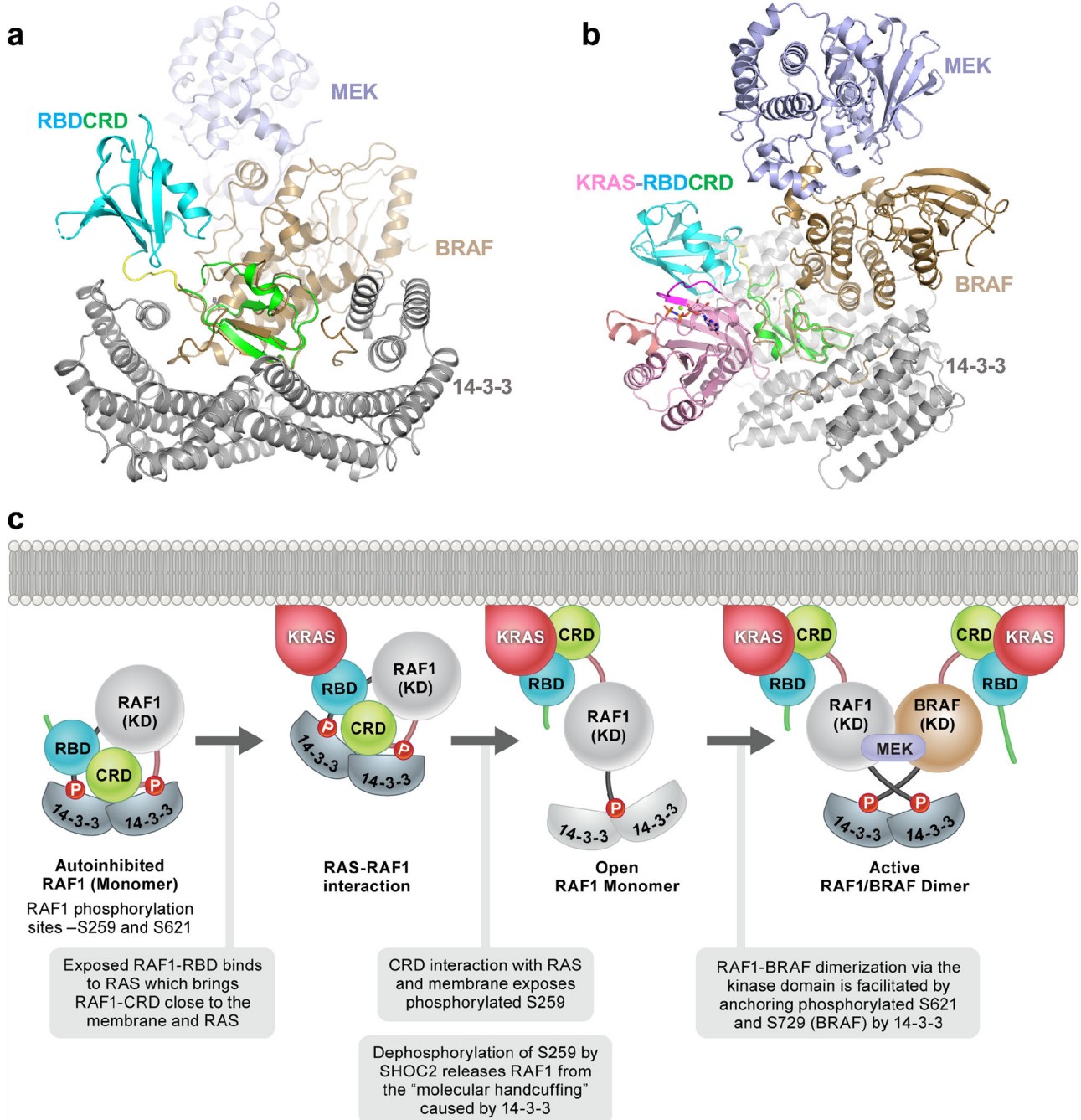

**Fig. 8 Model of RAS-mediated RAF activation using structural insights obtained from KRAS-RBDCRD and MEK-BRAF-14-3-3 structures. a** Structural superposition of KRAS-RBDCRD and the cryoEM structure of MEK-BRAF-14-3-3 (PDB 6NYB). Structures were aligned by their CRD domains. KRAS is hidden in this panel in order to show the alignment of CRD and the placement of RBD clearly. RAF1(RBDCRD) has the same color-coding scheme as used in Fig. 1c, while MEK, BRAF, and 14-3-3 are colored light blue, brown, and gray, respectively. **b** Same as **a** but at a different angle. KRAS is shown to illustrate it has minimal contacts with 14-3-3 in our structural alignment based on CRD. **c** Schematic diagram showing KRAS first interacts with autoinhibited RAF via RBD and then CRD to bring RAF close to the membrane. This results in CRD coming out of the autoinhibited conformation and interacting with the membrane and KRAS, and in turn exposing the phosphorylated serine present in the CR2 region. Dephosphorylation of S259 (RAF1) by SHOC2 complex results in an active RAF1 monomer, where 14-3-3 binds to only one phosphorylated serine located at the C-terminal end. Homo-dimerization or hetero-dimerization of KRAS-RAF1 with an active RAF monomer containing either RAF1 or BRAF results in the formation of an active dimerized RAF complex. RAF1 has been shown to prefer hetero-dimerization with BRAF which is bound to MEK for initiating the activation cascade.

KRAS4b protein. Avi-tagged clones were generated with an upstream TEV site (ENLYFQ) followed by the Avi tag sequence (GLNDIFEAQKIEWHEG) and the KRAS4b(1–169) sequence. In all cases, final protein expression constructs were generated by Gateway LR recombination into pDest-566 (Addgene #11517), an *Escherichia coli* T7-based expression vector based on pET42 and incorporating hexa-histidine (His6) and maltose-binding protein (MBP) tags at the N-terminus of the protein. Inactive human MEK1 (MAP2K1) with K97R mutation was synthesized by ATUM as an *E. coli* optimized Gateway Entry clone containing an upstream strong *E. coli* ribosome-binding site, and a downstream Avi-TEV-His6 fusion tag GLNDIFEAQKIEWHEGENLYFQGHHHHHH). Expression constructs were generated by Gateway LR recombination into pDest-521, a pET21 variant vector with a T7 promoter, and no fusion tags.

*Protein expression.* All RAF1(RBDCRD) proteins were expressed using the protocols outlined previously for RAF1 (52–188)[42]. KRAS, Avi-KRAS, RAF1(RBD), and MAP2K1 K97R-Avi-TEV-His6 proteins were expressed following protocols described in Taylor et al. (Dynamite media protocol, 16 °C induction)[41]. Essentially, an overnight 37 °C culture (non-inducing MDAG-135 medium) of the *E. coli* strain harboring the expression plasmid of interest was used as seed culture to inoculate (2% v/v) expression-scale cultures of Dynamite medium. The expression culture was grown at 37 °C until OD$_{600}$ reached 6–8, protein expression was induced with 0.5 mM IPTG, the culture was incubated at 16 °C for 18–20 h, and the cells were harvested by centrifugation.

*Protein purification.* All RAF1(RBDCRD) proteins were purified following the protocols described in Lakshman et al.[42], whereas KRAS, RAF1(52–131), and MAP2K1 K97R-Avi-TEV-His6 proteins were purified as outlined for KRAS4b (1–169) in Kopra et al.[43] Briefly, the expressed proteins of the form His6-MBP-TEV-target were purified from clarified lysates by IMAC, treated with His6-TEV protease to release the target protein, and the target protein separated from other components of the TEV protease reaction by the second round of IMAC. Proteins were further purified by gel-filtration chromatography in a buffer containing 20 mM HEPES, pH 7.3, 150 mM NaCl, 2 mM MgCl$_2$ (GTPases only), and 1 mM TCEP. The peak fractions containing pure protein were pooled, flash-frozen in liquid nitrogen, and stored at −80 °C.

**Crystallization and data collection.** Purified GDP-bound KRAS proteins were exchanged to GMPPNP-bound forms using the procedure reported earlier[44]. Wild type and oncogenic mutants (G12V, G13D, and Q61R) of KRAS were used for the crystallization experiments. The GMPPNP-bound KRAS proteins were mixed with RAF1(RBDCRD) or RAF1(RBD) in a 1:1.2 stoichiometric ratio and incubated for approximately 30–60 min. The protein–protein complex was then passed through a size-exclusion column (Superdex75, 26/60, GE Healthcare) to remove unbound proteins. Crystallization screening was carried out at 20 °C using the sitting-drop vapor diffusion method by mixing purified protein–protein complex (~9 mg/ml) with an equal volume of reservoir solution. Crystallization hits obtained from initial screening were optimized to improve the diffraction quality by systematically varying the pH, individual component concentrations, and the presence of additive and detergents (Supplementary Table 1). KRAS constructs with C118S mutation was also used to improve the resolution during screening and optimization as this mutation presumably reduces inadvertent cysteine oxidation during crystallization. Optimized crystals were harvested for data collection and cryoprotected with 25% (v/v) glycerol solution before being flash-frozen in liquid nitrogen. Diffraction data sets were collected at 24-ID-C/E beamlines at the Advanced Photon Source (APS), Argonne National Laboratory. The crystals of KRAS and KRAS-C118S complexed with RAF1(RBDCRD) belong to different space groups and diffracted to a resolution of 2.50 and 1.95 Å, respectively. Crystals of WT KRAS with RAF1(RBD) diffracted to a resolution of 1.4 Å, and oncogenic mutants of KRAS with RAF1(RBDCRD) diffracted to a resolution of 2.11–2.87 Å. Crystallographic datasets were integrated and scaled using XDS[45]. Crystal parameters and data collection statistics are summarized in Table 1.

**Structure determination and analysis.** The structure of WT KRAS in complex with RAF1(RBD) was solved by molecular replacement using the program Phaser as implemented in the Phenix suite of programs[46,47], with a protein-only structure of HRAS-RAF1(RBD) complex[19] as a search model (PDB: 4G0N). The structures of WT KRAS in complex with RAF1(RBDCRD) in crystal form I (1.95 Å) and II (2.5 Å) were solved using the protein-only structure of KRAS-RAF1(RBD) complex and the NMR structure of RAF1(CRD) (PDB: 1FAR)[20] as search models. The structures of oncogenic mutants (G12V, G13D, and Q61R) of KRAS in complex with RAF1(RBDCRD) were solved using the 2.5 Å WT KRAS-RAF1(RBDCRD) structure as the search model. The initial solution obtained from molecular replacement was refined using the program Phenix.refine within the Phenix suite of programs[46], and the resulting *Fo-Fc* map showed clear electron densities for the GMPPNP nucleotide, KRAS, and RBD/RBDCRD domains. The model was further improved using iterative cycles of manual model building in COOT[48] and refinement with Phenix.refine. GMPPNP nucleotide was placed in the nucleotide-binding pocket and followed by the addition of solvent molecules by the automatic water-picking algorithm in COOT. These solvent molecules were manually checked during model building until the final round of refinement was completed. Refinement statistics for the structures are summarized in Table 1. All of the structural figures were rendered in PyMOL (Schrödinger, LLC) or VMD[49] with secondary structural elements assigned using the DSSP server (http://swift.cmbi.ru.nl/gv/dssp). The amino acid sequence alignments were carried out using Clustal Omega[50], and the figures were produced using ESPript[51]. Crystallographic and structural analysis software support is provided by the SBGrid consortium[52].

**SPR measurements.** SPR binding experiments were collected on a Biacore T200 or S200 Instrument (GE Healthcare). Neutravidin (Pierce) was amine coupled to the carboxymethylated dextran surface of a CM5 sensor chip (GE Healthcare) using standard amine coupling chemistry. The CM5 chip surface was first activated with 0.1 M N-hydroxysuccinimide and 0.4 M N-ethyl-N′-(3-dimethylaminopropyl)

carbodiimide at a flow rate of 20 μl/min using 20 mM HEPES pH 7.4, 150 mM NaCl as the running buffer. Next, neutravidin was diluted to 20 μg/ml in 10 mM sodium acetate (pH 4.5) and injected on all four flow cells until a density of approximately 10,000 response units (RU) was immobilized. Activated amine groups were quenched with an injection of 1 M ethanolamine (pH 8.0). 100–400 RU of WT or mutant Avi-tagged KRAS (1–169) was captured on three flow cells in 20 mM HEPES pH 7.4, 150 mM NaCl, 5 mM MgCl$_2$, 1 mM TCEP, 5 μM GMPPNP, 0.01% tween 20 buffer. RAF1 RBD (52–131), CRD (136–188), RBDCRD (52–192), and RBDCRD mutant proteins were diluted from 20 to 0.05 μM in 20 mM HEPES pH 7.4, 150 mM NaCl, 5 mM MgCl$_2$, 1 mM TCEP, 5 μM GMPPNP, 0.01% tween 20 buffer, and injected over all flow cells at 30 μl/min. Complementary binding measurements using avi-CRD (136–188) and avi-RBDCRD (52–192) were captured as described above, except using a C1 chip. Farnesylated and methylated KRAS-FMe (1–185) and HVR-FMe (171–185) were diluted from 20 to 0.05 μM in 20 mM HEPES pH 7.4, 150 mM NaCl, 5 mM MgCl$_2$, 1 mM TCEP, 5 μM GMPPNP, 0.01% tween 20 buffer and injected over all the flow cells at 30 μl/min. Flow cell 1 was used for referencing and buffer injections were included for referencing purposes. SPR sensorgrams were normalized by the capture level of KRAS to allow direct comparison between different experimental runs. Steady-state levels of RBD or RBDCRD were recorded and fit with a 1:1 binding model using the Biacore Evaluation software, $K_D$ values are reported as the mean ± standard deviation.

**Immunoprecipitation/pulldown and kinase assays.** FLAG-RAF1 and Strep2-KRAS4b or empty vector mammalian expression plasmids were transiently expressed in 293T using JetPRIME transfection reagent according to the manufacturer protocol. Thirty-six hours later cells were washed with PBS and lysed on ice in 25 mM Tris pH 7.5, 150 mM NaCl, 1% Triton X-100, 5 mM MgCl$_2$, 1 mM DTT, EDTA-free protease inhibitors (Roche) and phosphatase inhibitor cocktail (Millipore-Sigma). After centrifugation to clear the lysates, anti-FLAG M2 (Millipore-Sigma) was used to isolate tagged-protein complexes by rotating for 2 h at 4 °C. Beads were washed with 25 mM Tris pH 7.5, 150 mM NaCl, 1% Triton X-100, 5 mM MgCl$_2$. A portion of the beads was analyzed by SDS-PAGE and Western blot, where complexes were visualized using antibodies to phosphorylated, tagged or endogenous proteins, and DyLight-conjugated secondary antibodies suitable for Li-COR Odyssey scanning. The remaining FLAG beads were incubated with 150 ng/μl 3× FLAG peptide (Millipore-Sigma) in kinase assay buffer (100 mM Tris pH 7.5, 150 mM NaCl, 5 mM MgCl$_2$, 1 mM DTT) for 10 min with agitation at 22 °C to elute proteins. The supernatant was collected, and the elution was repeated twice, followed by one wash with buffer without peptide to remove traces of eluate from the beads. To perform the kinase assay, FLAG-RAF1 eluate was serially diluted with kinase assay buffer and combined with recombinant MEK1-K97R-Avi (final [10 ng/μl]) and 1 mM ATP. Reactions were agitated at 22 °C for 20 min and were stopped with the addition of 4X NuPAGE LDS (ThermoFisher). A fraction of each assay sample was analyzed by Western blot as above. Quantification of band intensity was performed with ImageStudio Lite (Li-COR) software, and error bars represent S.D. of 3 independent experiments, except for the previously characterized RAF1-R89L which was repeated twice.

**Antibodies.** Avi-tag—GenScript A01738 (1:5000); MEK1—SCBT sc-6250 (1:500); FLAG—Sigma F7425 (1:4000); MEK P-S217/221—Cell Signaling 9121 (1:2000); RAF1—BD Transduction Laboratories 610152 (1:2000); NWSHPQFEK (Strep-tag) —GenScript A00626 (1:5000).

**Quantification of RAS orientation and membrane occlusion.** The orientation of KRAS with respect to the membrane surface is defined by two angles: the tilt angle, which separates the long axis of helix α5 from the bilayer normal, and the rotation angle, which defines which parts of KRAS are brought toward the membrane via tilting[8,28] (Supplementary Fig. 8a, b). At tilt = 0°, KRAS helix α5 is perpendicular to the membrane and all rotation angles are equivalent. At tilt = 90°, KRAS helix α5 is parallel to the membrane surface and angles of rotation = −15°, −85°, and 105° dispose KRAS helices α4/α5, helices α3/α4, and β strands β1–β3 toward the membrane, respectively (Supplementary Fig. 8b).

To predict the accessibility of KRAS binding sites, we constructed composite systems by aligning crystallographic KRAS with KRAS from all-atom MD simulations (ninety 5 μs simulations of GTP-bound KRAS, excluding the first 2 μs of each simulation). RAF1 proteins in these composite systems are then evaluated for steric overlap with lipids in the membrane associated with the simulated KRAS protein. The number of protein residues that clash with lipids (any protein atom within 0.1 nm of any lipid atom), $N_{res}^{clash}$, is used to assess the competence of the specific KRAS orientation to bind another protein in the absence of any lipidic accommodation. Orientation-specific ensemble averaging over a total of 270,000 simulation snapshots provides Boltzmann-weighted averages over the sampled displacement of the KRAS G-domain from the center of mass of the lipid bilayer along its normal, $d_{Gz}$.

**Modeling KRAS-RBDCRD complex at the membrane.** The KRAS-RBDCRD crystal structure was aligned such that the position and orientation of the KRAS G-domain matched configurations commonly observed in all-atom MD simulations

of KRAS on 7:3 POPC:POPS bilayers[28]. Quantitative definition of KRAS orientation is outlined in Supplementary Fig. 8a, b, and common orientations from simulation and experiment are shown in Supplementary Fig. 8c. Representative KRAS orientations from simulation were selected to maximize their similarity to the orientational centroid of that cluster and the average orientation-specific (±5° tilt and rotation) displacement of the KRAS G-domain from the center of mass of the lipid bilayer along its normal, $d_{Gz}$. KRAS fitting involves Cα atoms of KRAS G-domain residues Y4-N26, Y40-L56, and D69-H166 (selected to omit the relatively flexible N-terminus and switch regions of KRAS).

**Reporting summary**. Further information on research design is available in the Nature Research Reporting Summary linked to this article.

## Data availability

The atomic coordinates and structure factors of the various complexes have been deposited in the Protein Data Bank and are available under accession numbers: 6XI7—KRAS in complex with RAF1(RBDCRD), crystal form I; 6XHB—KRAS in complex with RAF1(RBDCRD), crystal form II; 6VJJ—KRAS in complex with RAF1(RBD); 6XHA—KRAS-G12V in complex with RAF1(RBDCRD); 6XGV—KRAS-G13D in complex with RAF1(RBDCRD); and 6XGU—KRAS-Q61R in complex with RAF1(RBDCRD). Data supporting the findings of this manuscript are available from the corresponding authors upon request. Source data are provided with this paper.

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

## Acknowledgements

We thank Bill Gillette, Jennifer Mehalko, Rosemilia Reyes, Vanessa Wall, Jose Sanchez Hernandez, Nitya Ramakrishnan, Allison Champagne, Peter Frank, Min Hong, Taylor Lohneis, Shelley Perkins, Stephanie Widmeyer, Matt Drew, Peter Frank, and Kelly Snead of the Protein Expression Laboratory (Frederick National Laboratory for Cancer Research) for their help in cloning, expressing, and purifying recombinant proteins. We are grateful to the staff of 24-ID-C/E beamline at the Advanced Photon Source, Argonne National Laboratory, for their help with data collection. Part of this work is based on research conducted at the Northeastern Collaborative Access Team beamlines, which are funded by the National Institute of General Medical Sciences from the National Institutes of Health Grant (P30 GM124165). The Eiger 16M detector on 24-ID-E is funded by a NIH-ORIP HEI grant (S10OD021527). This research used resources of the Advanced Photon Source, a US Department of Energy (DOE) Office of Science User Facility operated for the DOE Office of Science by Argonne National Laboratory under Contract DE-AC02-06CH11357. This project was funded in part with federal funds from the National Cancer Institute, National Institutes of Health Contract HHSN261200800001E. The content of this publication does not necessarily reflect the views or policies of the Department of Health and Human Services, and the mention of trade names, commercial products, or organizations does not imply endorsement by the US Government.

## Author contributions

T.H.T., A.H.C., and S.D. carried out structural work under the supervision of D.K.S.; L.C.Y. performed the immunoprecipitation/pulldown and kinase assays under the supervision of F.M.; L.B. carried out SPR experiments under the supervision of A.G.S.; and S.M., T.T., J.P.D., and D.E. prepared clones and recombinant proteins. C.N. carried out modeling studies. D.K.S., D.V.N., and F.M. provided resources and contributed to the analysis. T.H.T., A.H.C., F.M., and D.K.S. wrote the manuscript with inputs from all co-authors. D.K.S. supervised and coordinated the overall project.

## Funding

## Competing interests

The authors, except F.M., declare no competing interests. F.M. is a consultant for the following companies: Aduro Biotech, Amgen, Daiichi, Ideaya Biosciences, Kura Oncology, Leidos Biomedical Research, PellePharm, Pfizer, PMV Pharma, Portola Pharmaceuticals, and Quanta Therapeutics; has received research grants from Daiichi; is a recipient of funded research from Gilead Sciences; is a consultant and cofounder for the following companies (with ownership interest, including stock options): BridgeBio Pharma, DNAtrix, Olema Pharmaceuticals, and Quartz.
