## [Peer Review File · Nature Communications]

REVIEWER COMMENTS

Reviewer #1 (Remarks to the Author):

Tran, Chan and Young et al.: KRAS interaction with RAF1 RAS-binding domain and cysteine-rich domain provides insights into RAS-mediated RAF activation

Tran, Chan and Young et al. present a manuscript describing the first structure of a Ras protein in complex with the tandem RAF RBD-CRD domains. Contribution of CRD of RAF to Ras binding has been an open question for a couple of decades, and detailed structural and biochemical analysis in this manuscript shows that the interaction is primarily driven by the Ras-RBD interface, with further limited affinity provided by the Ras-CRD interaction. The authors claim that the Ras-CRD interaction is important for RAF kinase activity. Finally, the structure is used to evaluate existing NMR based models of Raf-RBD-CRD association with the membrane, showing that the structure as solved is compatible with Ras activation of RAF at the membrane.

This manuscript is generally well presented. Structural biology is well performed, and mutagenesis supports the structural conclusions. Structural and other biophysical data described here will be of interest to those in the field seeking to increase understanding of molecular mechanisms of RAF activation of Ras, which continues to be an open question. However, there are significant concerns that need to be addressed and the manuscript improved further to be suitable for publication in Nature Communications. Without those changes, although the data is of interest to the Ras signaling field, it does not hold enough broad appeal for Nature Communications and may be better suited to a journal that is more structure focused. The major insight of this article is the interaction surface of KRas and CRD. The functional consequence of this interaction and implications thereof for signaling are limited.

Major comments:

(1) Kinase activity of RAF and influence of CRD-KRas interaction

The kinase assay described beginning at line 305 is difficult to interpret. Is the purpose of this section to show that either (a) the Ras-CRD interaction is important for converting “autoinhibited Raf:Mek:1433 complex” to the “active dimer of RAF:MEK:1433”, or (b), that the Ras-CRD interaction is important for obtaining full kinase activity once the active dimeric state of Raf has been obtained? In either case, it does not seem that these conclusions could be drawn from the data presented, showing only small differences in effective RAF activity. Clarification of exactly what is being shown, and further experiments to support the conclusions would be helpful.

One interpretation of the experiment would be to expect that there are variable ratios of autoinhibited-CRAF:active-CRAF in the samples as purified, depending on the amount of specific RAF mutant converted into the active form by the constitutively active Ras. This is then expected to affect net kinase activity. Since it is known that there is a ~500-fold activity between the monomeric inactive and dimeric active forms of RAF, and a small change in the ratio could affect the net observed activity significantly. Alternatively, the mutations themselves may affect RAF kinase activity. As it stands, it is impossible to determine whether these mutations affect autoinhibited:active CRAF ratios, or whether they affect CRAF kinase activity once activated.

Overall, the differences in the measured RAF activity are small, and it is tough to tie it back to relevance of each single mutation.

Some other more minor points relating to the kinase activity section are:

(a) How much RAF was active vs inactive?

(b) RAF was Co-purified with constitutively active KRAS. But how did you know that all of the RAF was bound to RAS and active?

(c) For the RAF mutants which don't bind KRAS very well, why is the kinase activity so high?

(d) Please consider biological replicates in addition to technical replicates. It is tough to interpret which differences are due to prep to prep variation versus real differences in activity as a result of the mutations.

(e) 40% of Rasopathy mutations occur in the CRD. Does your mutagenesis work cover any of these, and are they explained by binding affinity differences? Or are they likely to be more complicated?

(2) Structural Analysis

There is a significant number of literature about a potential Ras dimer forming at the membrane, upon activation of the pathway. A large number of KRas crystal structures across different crystal forms reveal an interaction surface formed by $\alpha 4$ and $\alpha 5$ helices, resulting in a symmetric Ras dimer. A 2018 paper in Cell from Ken Westover group also revealed that a D154Q mutation at this dimer interface resulted in disruption of signaling and a double mutant D154Q, R161E mutation restored signaling by KRas. Even though there appear to be two schools of thought on the potential Ras dimer being relevant, it may be worthwhile to perform further analysis of the two crystal forms for the KRas:RBD-CRD structures to see if the Ras molecules form a similar dimer interacting via the $\alpha 4$ and $\alpha 5$ helices. This would also feed into the analysis of the NMR data and MD simulations to generate a model for the membrane bound KRas:RAF-RBD-CRD complex. In the current suggested model, the $\alpha 4$, $\alpha 5$ surface is interacting with the membrane, and occluded from the previously suggested dimer interface. The manuscript would benefit from structural analysis and hypothesis generation on this end.

Crystallography:

(a) $I/\sigma(I)$ is very high for the highest resolution shell for crystal form II. It is also high for other structures as well. Looks like the data was truncated using R-meas and CC1/2 cutoff. Based on the data shown (high redundancy) especially for Crystal form II, it is very likely that there is radiation damage in the crystal as the data collection progressed and data should be reprocessed using less number of frames (initial frames), to limit the redundancy. This will likely allow for a higher resolution and cleaner dataset (if the data were collected to high enough resolution). This may work well for all the datasets too.

(b) R-merge is not useful unless datasets are being merged from different crystals.

(c) Since the CRD structure has been obtained for the first time by crystallography (and high resolution) please comment on the occupancy of the CRD domain in the crystal structure and B-factors. Since the improvement in affinity beyond just the RBD is small due to the addition of CRD, it is likely that the interaction is not very stable. It would be important to comment on that and have a supplementary figure showing electron density at the interface of CRD/KRas.

(3) Other points:

1) Typo in figure 5a, R49A should be R59A

2) The HVR region in the KRas4b-PDE6D complex solved by the authors previously is helical and ordered, but in the models shown here, the HVR region is unfolded. Were the MD simulations started with the helical HVR model and resulted in the unfolded state upon simulation?

Reviewer #2 (Remarks to the Author):

The authors describe a series of crystal structures of KRAS in complex with N-terminal CRAF fragments containing either the RBD domain or RBD+CRD domains. In two different crystal forms, they find that both the RBD and CRD domains make extensive interactions with KRAS. RBD interactions are essentially the same as that seen before in similar structures of RAF RBD domains bound to HRAS and Rap1 (and also in the RAS/RBD crystal structure described here). The news here is the extensive interaction of the CRD, as well as interactions of the short linker connecting RBD and CRD domains, which also contacts RAS. The authors present extensive structure/function work that shows that although mutation of the CRD/RAS interface has only modest effects on the affinity of the RAS/RBD+CRD interaction, disruption of the interface nevertheless diminishes RAF1 kinase activity in IP-kinase experiments.

Overall, this a solid study that represents an important step forward in our understanding of RAS-driven recruitment and activation of Raf. The crystal structures are at moderate to high resolution and appear to have been carefully refined and interpreted. The work should receive a high priority for publication in Nature Communications with attention to the following points:

1. This is a long hard read - paper would benefit from shortening.
2. A recent NMR study of KRAS interactions with the RAF RBD+CRD region is cited here, but rather obliquely. Fang et al. studied the interactions of Ras with RBD+CRD both in solution and with KRAS anchored in membrane nanodiscs. Though no structure was deposited for the solution-state interactions, the authors note chemical shift perturbation of the Ras $\alpha 5$ helix (which is consistent with the crystal structures described here). Fang et al. provide models for two states of the complex in nanodiscs (as derived from NMR data). I would like to see the structure described here compared directly to the NMR-derived models. It looks to me that they are rather different with respect to the CRD? (Current Supplemental Fig. 6 is not helpful in this regard - best to show the structures/models side-by-side or superposed.)
3. Is it possible that the conformation observed in the present crystal structure is relevant to recruitment of RAF by Ras, but not the "final" membrane-associated conformation?
4. It is not clear to me from figures & methods that the RAF activity measured in Fig. 5 is necessarily dependent on Ras binding, as implied. The "Empty" control in Fig. 5b appears to be a no-RAF control, not a no-RAS control. I presume that the corresponding lane in the experiment in 5d is a no-Ras control, if so it gives some reassurance that the authors are looking at RAS-dependent activity. But addition of an active RAS mutant that can't bind RAF (Q61L/R89L) would be a welcome control in both experiments.

-Michael Eck

Reviewer #3 (Remarks to the Author):

In this manuscript, Tran et al. report the crystal structures of the GTP-bound forms of wild-type and oncogenic KRAS in complex with the RAS-binding domain (RBD) and the cysteine-rich domain (CRD) of RAF-1. It is highly evaluated that the authors for the first time succeed in determination of the crystal structure of the KRAS-RAF1(RBDCRD) complex and clarification of the molecular mechanism underlying RAS-CRD interaction, whose significance in full activation of RAF has been shown by a number of past biochemical and cell biological studies, although the results provide only a limited clue to the overall mechanism for RAF activation by RAS at the plasma membranes, which is known to be extremely complex. The results regarding the acquired crystal structures are well presented and their interpretation seems fair. However, the use of E. coli-produced unmodified form of RAS for the complex formation substantially depreciates the value of the results and their implication

because almost all the papers studying the interaction of RAS with the isolated CRD have shown that farnesylation of Ras at its C terminus is necessary for this interaction (for example, references 8, 12, 14 and 18 with an exception of 13 only in this manuscript. Also, there are more papers supporting the farnesylation requirements such as Luo et. al. Mol. Cell. Biol. 17: 46-53, 1997.). Therefore, the actual mode of RAS-CRD interaction might be somewhat different from that reported here. If the authors have arguments against the farnesylation requirements, they must include them into the manuscript.

In the same line, the use of unmodified KRAS, not farnesylated KRAS, in SPR experiments are possibly inappropriate for the determination of real binding affinities of RAF1(RBDCRD) and KRAS. Also, the use of RAF1(RBDCRD) polypeptide in SPR experiments makes it difficult to observe the effects of mutations of KRAS and CRD on the binding affinity because the CRD binding is masked by the predominant binding at RBD (actually modest effects are observed). Use of isolated CRD polypeptide would be recommended for this purpose. Moreover, it is puzzling that the sensorgrams of KRAS-RAF1 binding presented in Supplementary Figures 1, 2 and 4 are all square shaped, which appears to have forced the authors to determine the Kd values by an unusual method measuring the dose-dependent increase of the binding signal, not by a standard method measuring binding kinetics (ka and kd), in a Biacore machine. The authors must explain the reason why the sensorgrams are all square shaped and why they don't use the standard method of SPR measurement.

Point-by-point response to Reviewer's comments

General comments:

We appreciate the extensive and thoughtful suggestions from the reviewers. In the revised manuscript, we have addressed the concerns of all three reviewers. The reviewers' feedback has led to a significantly improved manuscript. In the sections that follow, we have provided point-by-point responses (in blue) to the reviewers' critiques.

Reviewer #1 (Remarks to the Author):

Tran, Chan and Young et al.: KRAS interaction with RAF1 RAS-binding domain and cysteine-rich domain provides insights into RAS-mediated RAF activation

Tran, Chan and Young et al. present a manuscript describing the first structure of a Ras protein in complex with the tandem RAF RBD-CRD domains. Contribution of CRD of RAF to Ras binding has been an open question for a couple of decades, and detailed structural and biochemical analysis in this manuscript shows that the interaction is primarily driven by the Ras-RBD interface, with further limited affinity provided by the Ras-CRD interaction. The authors claim that the Ras-CRD interaction is important for RAF kinase activity. Finally, the structure is used to evaluate existing NMR based models of Raf-RBD-CRD association with the membrane, showing that the structure as solved is compatible with Ras activation of RAF at the membrane.

This manuscript is generally well presented. Structural biology is well performed, and mutagenesis supports the structural conclusions. Structural and other biophysical data described here will be of interest to those in the field seeking to increase understanding of molecular mechanisms of RAF activation of Ras, which continues to be an open question. However, there are significant concerns that need to be addressed and the manuscript improved further to be suitable for publication in Nature Communications. Without those changes, although the data is of interest to the Ras signaling field, it does not hold enough broad appeal for Nature Communications and may be better suited to a journal that is more structure focused. The major insight of this article is the interaction surface of KRas and CRD. The functional consequence of this interaction and implications thereof for signaling are limited.

We thank the reviewer for finding our manuscript "well presented" and recognizing that the structural and biophysical data will be of interest to those in the field seeking to increase understanding of molecular mechanisms of RAF activation of Ras. In the revised version of the manuscript, we have incorporated most of his/her suggestions.

Major comments:

(1) Kinase activity of RAF and influence of CRD-KRas interaction

The kinase assay described beginning at line 305 is difficult to interpret. Is the purpose of this section to show that either (a) the Ras-CRD interaction is important for converting "autoinhibited Raf:Mek:1433 complex" to the "active dimer of RAF:MEK:1433", or (b), that the Ras-CRD interaction is important for obtaining full kinase activity once the active dimeric state of Raf has been obtained? In either case, it does not seem that these conclusions could be drawn from the

data presented, showing only small differences in effective RAF activity. Clarification of exactly what is being shown, and further experiments to support the conclusions would be helpful.

One interpretation of the experiment would be to expect that there are variable ratios of autoinhibited-CRAF:active-CRAF in the samples as purified, depending on the amount of specific RAF mutant converted into the active form by the constitutively active Ras. This is then expected to affect net kinase activity. Since it is known that there is a ~500-fold activity between the monomeric inactive and dimeric active forms of RAF, and a small change in the ratio could affect the net observed activity significantly. Alternatively, the mutations themselves may affect RAF kinase activity. As it stands, it is impossible to determine whether these mutations affect autoinhibited:active CRAF ratios, or whether they affect CRAF kinase activity once activated. Overall, the differences in the measured RAF activity are small, and it is tough to tie it back to relevance of each single mutation.

Reply: The purpose of the kinase section was to simply show that single amino acid substitutions in the CRD can affect RAS-RAF interaction and RAF activation. For this, we used an assay format that has been used previously to identify residues that contribute to activation. We purified and analyzed the activity of 14 RAF mutants and 7 KRAS mutants, and repeated the whole experiment twice, as suggested, and included the well-characterized R89L (RAF1) mutant. In this assay format, most of the transfected RAF1 binds to activated KRAS, but we cannot distinguish between sub-populations that are in different states of activation. We have updated Figure 5 and related Supplementary Figure 7 with new results and modified the text for this part to make it easier to interpret. Analysis of RAF kinase purified from mammalian cells following activation by KRAS, by our lab and by others, reveals phosphorylation at several sites, as well as association with 14-3-3 and MEK. It would be of interest to determine the proportion of RAF kinase molecules in different states, but we feel this would be beyond the scope of this study.

Some other more minor points relating to the kinase activity section are:

(a) How much RAF was active vs inactive?

Reply: As we noted in our reply to the above question 1, in this assay format, most of the transfected RAF1 binds to activated KRAS, but we cannot distinguish between sub-populations that are in different states of activation. We agree that it would be of interest to determine the proportion of RAF kinase molecules in different states, but we feel this would be beyond the scope of this study.

(b) RAF was Co-purified with constitutively active KRAS. But how did you know that all of the RAF was bound to RAS and active?

Reply: Please see our reply to the above question 1.

(c) For the RAF mutants which don't bind KRAS very well, why is the kinase activity so high?

Reply: In this assay format, KRAS is highly expressed and active. We believe that under these conditions, even RAF1 RBD mutants that bind poorly can be activated as long as the CRD is intact. For example, even the R89L mutant has detectable kinase activity, as shown in Figure 5b.

(d) Please consider biological replicates in addition to technical replicates. It is tough to interpret which differences are due to prep to prep variation versus real differences in activity as a result of the mutations.

Reply: We have repeated analysis of the RAF and RAS mutants twice, as full biological replicates, now including R89L (RAF1) as requested.

(e) 40% of Rasopathy mutations occur in the CRD. Does your mutagenesis work cover any of these, and are they explained by binding affinity differences? Or are they likely to be more complicated?

Reply: Unfortunately, we are unable to comment on the relevance of CRD mutations to Rasopathies. BRAF-CRD mutations have been shown to cause RASopathy, and none of the relevant residues are conserved between BRAF and CRAF. Since we used CRAF/RAF1 for the structural and functional studies, we cannot explain binding affinity differences for BRAF-CRD mutations involved in RASopathy.

Mutations in RAF1 that cause Noonan Syndrome (RASopathy) are typically around residue S259, or in the kinase domain (please see attached data from UniProt). Since the kinase domain is not included in our crystal structure, we are unable to comment on the relevance of Rasopathy mutations in this study.

Feature key	Position(s)	Description	Actions	Graphical view
Natural variant ¹ (VAR_037807)	256	R → S in NS5. 1 Publication Corresponds to variant dbSNP:rs397516826	Ensembl , ClinVar .	
Natural variant ¹ (VAR_037808)	257	S → L in NS5 and LPRD2; shows in vitro greater kinase activity and enhanced ERK activation than wild-type. 2 Publications Corresponds to variant dbSNP:rs80338796	Ensembl , ClinVar .	
Natural variant ¹ (VAR_037809)	259	S → F in NS5. 1 Publication Corresponds to variant dbSNP:rs397516827	Ensembl , ClinVar .	
Natural variant ¹ (VAR_037811)	260	T → R in NS5. 1 Publication		
Natural variant ¹ (VAR_037812)	261	P → A in NS5; shows in vitro greater kinase activity and enhanced MAPK1 activation than wild-type. 1 Publication Corresponds to variant dbSNP:rs121434594	Ensembl , ClinVar .	
Natural variant ¹ (VAR_037813)	261	P → L in NS5; shows greater kinase activity and enhanced MAPK1 activation than wild-type. 1 Publication Corresponds to variant dbSNP:rs397516828	Ensembl , ClinVar .	
Natural variant ¹ (VAR_037814)	261	P → S in NS5; shows in vitro greater kinase activity and enhanced MAPK1 activation than wild-type. 3 Publications Corresponds to variant dbSNP:rs121434594	Ensembl , ClinVar .	
Natural variant ¹ (VAR_037815)	263	V → A in NS5; shows in vitro greater kinase activity and enhanced MAPK1 activation than wild-type. 1 Publication Corresponds to variant dbSNP:rs397516830	Ensembl , ClinVar .	
Natural variant ¹ (VAR_037816)	486	D → G in NS5. 1 Publication Corresponds to variant dbSNP:rs397516815	Ensembl .	
Natural variant ¹ (VAR_037817)	486	D → N in NS5; has reduced or absent kinase activity. 1 Publication Corresponds to variant dbSNP:rs80338798	Ensembl .	
Natural variant ¹ (VAR_037818)	491	T → I in NS5; has reduced or absent kinase activity. 1 Publication Corresponds to variant dbSNP:rs80338799	Ensembl .	
Natural variant ¹ (VAR_037819)	491	T → R in NS5. 1 Publication Corresponds to variant dbSNP:rs80338799	Ensembl .	
Natural variant ¹ (VAR_037820)	612	S → T in NS5. 1 Publication Corresponds to variant dbSNP:rs1448392469	Ensembl .	
Natural variant ¹ (VAR_037821)	613	L → V in NS5 and LPRD2; shows in vitro greater kinase activity and enhanced MAPK1 activation than wild-type. 2 Publications Corresponds to variant dbSNP:rs80338797	Ensembl .	

(2) Structural Analysis

There is a significant number of literature about a potential Ras dimer forming at the membrane, upon activation of the pathway. A large number of KRas crystal structures across different crystal forms reveal an interaction surface formed by $\alpha 4$ and $\alpha 5$ helices, resulting in a symmetric Ras dimer. A 2018 paper in Cell from Ken Westover group also revealed that a D154Q mutation at this dimer interface resulted in disruption of signaling and a double mutant D154Q, R161E mutation restored signaling by KRas. Even though there appear to be two schools of thought on the potential Ras dimer being relevant, it may be worthwhile to perform further analysis of the two crystal forms for the KRas:RBD-CRD structures to see if the Ras molecules form a similar dimer interacting via the $\alpha 4$ and $\alpha 5$ helices. This would also feed into the analysis of the NMR data and MD simulations to generate a model for the membrane bound KRas:RAF-RBD-CRD complex. In the current suggested model, the $\alpha 4$, $\alpha 5$ surface is interacting with the membrane, and occluded from the previously suggested dimer interface. The manuscript would benefit from structural analysis and hypothesis generation on this end.

Reply: We thank the Reviewer for pointing out the simultaneous incompatibility of our proposed model of the membrane-bound KRAS:RAF-RBD-CRD complex with the previously proposed symmetric RAS dimer interface formed by $\alpha 4/\alpha 5$ helices.

Using the symmetry-related $\alpha 4$ and $\alpha 5$ helices of HRAS in the crystal structure of HRAS:RAF1-RBD (PDB: 4G0N) as a guide (**Fig. R1a**), we observe similar packing in crystal form I of KRAS-RBDCRD (**Fig. R1b**). However, we do not observe similar RAS dimers in crystal form II of KRAS-RBDCRD (**Fig. R1c**). Nor do we observe it for KRAS-RBD (**Fig. R1d**). In fact, this mode of association is only observed once among a total of 11 KRAS pairs that could be interpreted as dimers based on a minimum intermolecular distance ($<4 \text{ \AA}$) between the KRAS-RAF complexes present in the asymmetric units among these later three crystal structures (**Table R1**). Therefore, we have chosen to put forth a model of membrane-associated KRAS:RAF-RBDCRD complex based on CRD residues that have been shown to interact with the membrane and three highly populated orientations (**Fig. 7c**) of KRAS on a membrane (containing 30% PS) rather than the proposed symmetric $\alpha 4/\alpha 5$ RAS dimer interface with CRD residues incompatible with membrane insertion. To clarify this incompatibility, we have added the following text to the end of the Results section on page 13:

“However, because this model places KRAS helices $\alpha 4$ and $\alpha 5$ in membrane contact, it is not simultaneously compatible with RAS dimerization at this interface as proposed by Fang *et al* in their State B KRAS-RBDCRD model and others [Spencer-Smith *et al.*, *Nat. Chem. Biol.*, 2017, 13: 62-68] [Ambrogio *et al.*, *Cell*. 2018, 172:857-868].

Figure R1: Potential dimer interfaces that are most similar to the proposed symmetric $\alpha4/\alpha5$ RAS dimer interface. The similarity of potential dimers is evaluated by the root mean squared deviation (RMSD) of KRAS C_{α} atoms to the $\alpha4/\alpha5$ RAS dimer seen in the crystal structure of HRAS:RAF1(RBD) complex (PDB: 4G0N). For each crystal structure, the pair of KRAS molecules having the lowest RMSD for a potential dimer (minimum intermolecular distance <4 Å) is shown. KRAS images are orange and yellow, and KRAS residues in the $\alpha4/\alpha5$ interface (residues 125 to ~166) are black. (a) Definition of the symmetric $\alpha4/\alpha5$ RAS dimer interface is based on the crystal structure of HRAS:RAF1(RBD) complex (PDB: 4G0N). (b) The most similar potential dimer KRAS pair in crystal form I of KRAS-RBDCRD; RMSD = 2 Å. (c) The most similar potential dimer KRAS pair in crystal form II of KRAS-RBDCRD; RMSD = 51 Å. (d) The most similar potential dimer KRAS pair in KRAS-RBD; RMSD = 26 Å.

Table R1: Summary of crystal interfaces and their similarity to the proposed symmetric $\alpha4/\alpha5$ RAS dimer.

Structures	Number of RAS-RAF complexes with minimum distance < 4 Å to the RAS-RAF complex present in the asymmetric unit	Minimum KRAS C_{α} RMSD to the proposed symmetric $\alpha4/\alpha5$ RAS dimer interface (Å)
HRAS-RBD (PDB: 4G0N)	3	0
KRAS-RBDCRD form I	3	2
KRAS-RBDCRD form II	2	51
KRAS-RBD	6	26

Crystallography:

(a) $I/\sigma I$ is very high for the highest resolution shell for crystal form II. It is also high for other structures as well. Looks like the data was truncated using R-meas and CC1/2 cutoff. Based on the data shown (high redundancy) especially for Crystal form II, it is very likely that there is radiation damage in the crystal as the data collection progressed and data should be reprocessed using less number of frames (initial frames), to limit the redundancy. This will likely allow for a higher resolution and cleaner dataset (if the data were collected to high enough resolution). This may work well for all the datasets too.

Reply: We thank the reviewer for pointing this out. We agree that we were a little strict in deciding the final resolution of some of the crystallographic datasets presented in this manuscript. As per the reviewer's suggestions, we reprocessed three datasets that had relatively high $I/\sigma I$ in the highest resolution shell. We were able to extend the resolution of these three structures by 0.05-0.15 Ang, and this also resulted in a slightly improved electron density map.

(b) R-merge is not useful unless datasets are being merged from different crystals.

Reply: We agree with the reviewer's suggestions. We included the R-merge value in the crystallography table as per the journal guidelines. In the crystallography table, we have also included R-meas, which represent redundancy independent merging R-value.

(c) Since the CRD structure has been obtained for the first time by crystallography (and high resolution) please comment on the occupancy of the CRD domain in the crystal structure and B-factors. Since the improvement in affinity beyond just the RBD is small due to the addition of CRD, it is likely that the interaction is not very stable. It would be important to comment on that and have a supplementary figure showing electron density at the interface of CRD/KRAs.

Reply: The occupancy and the B-factors of residues present in CRD are similar to residues corresponding in RBD and KRAS. Only the linker region (residues 132-138) has a relatively high-b-factor compared to RBD and CRD. Even though the interaction between KRAS and RBDCRD is predominantly coming from the KRAS-RBD interface, there are nine hydrogen bonds at the KRAS-CRD interface, and we see a similar quality of electron density map at KRAS-RBD and KRAS-CRD interfaces. As suggested, we have included a new supplementary figure (Supplementary Fig. 4) showing electron density at CRD and KRAS interface.

(3) Other points:

1) Typo in figure 5a, R49A should be R59A.

Reply: Thank you for pointing this out. We have corrected this in the revised manuscript.

2) The HVR region in the KRAs4b-PDE6D complex solved by the authors previously is helical and ordered, but in the models shown here, the HVR region is unfolded. Were the MD simulations started with the helical HVR model and resulted in the unfolded state upon simulation?

Reply: In the KRAs4b-PDE6D complex structures solved in two different crystal forms, we observed the HVR region (169-179) as helical ordered in only one crystal form. In the second

crystal form, a part of the HVR region (173-179) was disordered likely due to inherent flexibility and lack of helical structure. To avoid any bias, the MD simulations were not started with a helical HVR model. These simulations originated as enhanced sampling (temperature replica exchange) simulations of lipid bilayers and the HVR alone (i.e., lacking the G domain entirely), initiated from an all-*trans* HVR. Those simulations were published as [Neale and Garcia, 2018, *J. Phys. Chem. B*, 122(44): 10086-10096]. In a subsequent simulation study, we attached KRAS G domains to representative HVR peptides from these HVR-only simulations and proceeded to simulate full-length KRAS molecules on lipid bilayers [Neale and Garcia, 2020, *Biophys. J.*, 118(5): 1129-1141]. Finally, representative models of full-length KRAS on lipid bilayers were extracted from these simulations and used in conjunction with the crystal structure of KRAS-RBDCRD to generate the models shown in this manuscript.

Reviewer #2 (Remarks to the Author):

The authors describe a series of crystal structures of KRAS in complex with N-terminal CRAF fragments containing either the RBD domain or RBD+CRD domains. In two different crystal forms, they find that both the RBD and CRD domains make extensive interactions with KRAS. RBD interactions are essentially the same as that seen before in similar structures of RAF RBD domains bound to HRAS and Rap1 (and also in the RAS/RBD crystal structure described here). The news here is the extensive interaction of the CRD, as well as interactions of the short linker connecting RBD and CRD domains, which also contacts RAS. The authors present extensive structure/function work that shows that although mutation of the CRD/RAS interface has only modest effects on the affinity of the RAS/RBD+CRD interaction, disruption of the interface nevertheless diminishes RAF1 kinase activity in IP-kinase experiments.

Overall, this a solid study that represents an important step forward in our understanding of RAS-driven recruitment and activation of Raf. The crystal structures are at moderate to high resolution and appear to have been carefully refined and interpreted. The work should receive a high priority for publication in Nature Communications with attention to the following points:

We thank the reviewer for finding our work “a solid study” and an important step forward in our understanding of RAS-mediated Raf activation. In the revised manuscript, we have incorporated suggestions made by this reviewer.

1. This is a long hard read - paper would benefit from shortening.

Reply: Based on the reviewer’s suggestion, we have shortened the Introduction, Results, and Discussion sections to keep the focus on the main findings of this study.

2. A recent NMR study of KRAS interactions with the RAF RBD+CRD region is cited here, but rather obliquely. Fang et al. studied the interactions of Ras with RBD+CRD both in solution and with KRAS anchored in membrane nanodiscs. Though no structure was deposited for the solution-state interactions, the authors note chemical shift perturbation of the Ras $\alpha 5$ helix (which is consistent with the crystal structures described here). Fang et al. provide models for two states of the complex in nanodiscs (as derived from NMR data). I would like to see the structure

described here compared directly to the NMR-derived models. It looks to me that they are rather different with respect to the CRD? (Current Supplemental Fig. 6 is not helpful in this regard - best to show the structures/models side-by-side or superposed.)

Reply: We thank the Reviewer for pointing out our failure to effectively compare our model of KRAS-RBDCRD membrane interaction to that of Fang *et al.* To address this, we have included a new Supplementary Figure 9 that makes this explicit comparison.

We have also added the main text to the results that refer to Supplementary Figure 9 as follows:

We note that our model of membrane-bound KRAS-RBDCRD shown in **Fig. 7c** is similar to the State A configuration proposed recently by Fang *et al.*⁶ based on an NMR data-driven model generated using docking software HADDOCK, excepting a 180° rotation of CRD about the approximate RBDCRD long axis (**Supplementary Fig. 8i**). Whereas the previous model placed CRD helix ¹⁷⁴EHCSTKV¹⁸⁰ away from KRAS, our crystal structures indicate that this helix makes direct contact with the KRAS G-domain (**Supplementary Fig. 9a, b**). This discrepancy of the CRD orientation could be due to insufficient NMR distance restraints on the CRD's β 1 and 3₁₀-helix (since many residues in these regions were broadened beyond detection), the presence of the nanodisc in the NMR experiment, and/or perhaps higher flexibility of CRD in solution in general as indicated by the many broadened peaks. Importantly, membrane interactions of CRD loop residues ¹⁴³RKTFLKLA¹⁵¹ and ¹⁵⁷KLLNGFR¹⁶⁴ are relatively unaffected when KRAS abuts the membrane with G domain helices 4 and 5 (**Supplementary Fig. 9c, d**).

Supplementary Figure 9: Influence of the KRAS-CRD interface on CRD-membrane contacts. (a, b) Structural comparison of KRAS-RBDCRD in (a) NMR data-driven HADDOCK model from Fang *et al.*⁶ and (b) crystal structure of KRAS-RBDCRD presented in this work. Note the ~180° rotation of the CRD about its long axis in panel a vs panel b. Helical CRD residues ¹⁷⁴EHCSTKV¹⁸⁰ are shown in black. (c) PDB: 6PTS model 0 from Fang *et al.*⁶. (d) Crystallographic KRAS-RBDCRD aligned on KRAS G-domain C α atoms from PDB: 6PTS model 0, showing lipids from the later model.

3. Is it possible that the conformation observed in the present crystal structure is relevant to recruitment of RAF by Ras, but not the "final" membrane-associated conformation?

Reply: The Reviewer makes an excellent point. Considering membrane-interacting CRD residues in the KRAS-RBDCRD structure are fully-exposed on the CRD surface, this could allow the RAS-RAF complex to interact with the membrane while the CRD is bound to KRAS. However, our results do not rule out that the final membrane-associated conformation of RAF could be different from the initial RAS-RAF complex. Keeping this in mind, we have added the following text to the end of the Results section on page 13:

Furthermore, the KRAS-RBDCRD complex may undergo further reorganization driven by membrane association [Li *et al.*, *ACS Cent Sci* 2018 4, 298-305].

4. It is not clear to me from figures & methods that the RAF activity measured in Fig. 5 is necessarily dependent on Ras binding, as implied. The "Empty" control in Fig. 5b appears to be a no-RAF control, not a no-RAS control. I presume that the corresponding lane in the experiment in 5d is a no-Ras control, if so it gives some reassurance that the authors are looking at RAS-dependent activity. But addition of an active RAS mutant that can't bind RAF (Q61L/R89L) would be a welcome control in both experiments.

Reply: As suggested by the reviewer, we have now included RAF1 (R89L) as a control in both experiments described in Fig. 5 and Supplementary Figure 7. We have now amended these two figures to include RAF1 only (no RAS) and RAF1-R89L as controls rather than a no-RAF control, demonstrating that kinase activity is RAS-dependent. The reviewer is correct in that the 1st bar in Fig. 5d is a RAF only, a no RAS control.

Reviewer #3 (Remarks to the Author):

In this manuscript, Tran et al. report the crystal structures of the GTP-bound forms of wild-type and oncogenic KRAS in complex with the RAS-binding domain (RBD) and the cysteine-rich domain (CRD) of RAF-1. It is highly evaluated that the authors for the first time succeed in determination of the crystal structure of the KRAS-RAF1(RBDCRD) complex and clarification of the molecular mechanism underlying RAS-CRD interaction, whose significance in full activation of RAF has been shown by a number of past biochemical and cell biological studies, although the results provide only a limited clue to the overall mechanism for RAF activation by RAS at the plasma membranes, which is known to be extremely complex. The results regarding the acquired crystal structures are well presented and their interpretation seems fair. However, the use of *E. coli*-produced unmodified form of RAS for the complex formation substantially depreciates the value of the results and their implication because almost all the papers studying the interaction of RAS with the isolated CRD have shown that farnesylation of Ras at its C terminus is necessary for this interaction (for example, references 8, 12, 14 and 18 with an exception of 13 only in this manuscript. Also, there are more papers supporting the farnesylation requirements such as Luo et al. *Mol. Cell. Biol.* 17: 46-53, 1997.). Therefore, the actual mode of RAS-CRD interaction might be somewhat different from that reported here.

We thank this reviewer for finding our work of high value and recognizing that the crystal structures are well presented, and interpretations are fair. In the revised version of the manuscript, we have

incorporated most of his/her suggestions.

If the authors have arguments against the farnesylation requirements, they must include them into the manuscript. In the same line, the use of unmodified KRAS, not farnesylated KRAS, in SPR experiments are possibly inappropriate for the determination of real binding affinities of RAF1(RBDCRD) and KRAS.

Reply: We used both full-length farnesylated and methylated KRAS (KRAS-FMe) produced in insect cells and non-processed G-domain of KRAS (residue 1-169) produced in *E. coli* for our structural and biophysical studies. Our efforts to crystallize full-length farnesylated KRAS with RAF1(RBDCRD) failed to give any crystallization hits. In contrast, we could obtain crystals of KRAS(G-domain) with RAF1(RBDCRD) in multiple crystal forms. This suggested that the farnesylated and methylated hypervariable region (HVR) of KRAS is likely to be flexible (not interacting with RAF1-CRD) and possibly hindering the crystallization process. As suggested, we carried out extensive SPR binding studies using fully processed KRAS-FMe, farnesylated HVR peptide (HVR-FMe), and non-processed KRAS(G-domain) with RAF1 CRD and RBDCRD. Overall, our results do not support interaction between farnesylated HVR and CRD as reported earlier, at least under the conditions used in our experiments. Also, considering farnesylated HVR is important for KRAS-membrane interaction, it is not clear why this would change when RAS interacts with RAF1-CRD. In the revised manuscript, we have included these results in a new Supplementary Figure 6 and the following text on page 9:

Previous studies have suggested that the farnesylated HRAS interacts with CRD in the low micromolar range^{12,14}. Our efforts to crystallize farnesylated and methylated full-length KRAS (KRAS-FMe) in complex with RAF1(RBDCRD) failed to give any crystallization hits. To investigate the role of the farnesyl group in the interaction between KRAS and RAF1 CRD/RBDCRD, we carried out SPR binding analysis using active fully processed KRAS-FMe and non-processed KRAS (G-domain) on the chip surface and flowed CRD and RBDCRD over them. RBDCRD bound with an equivalent affinity to non-processed KRAS and fully processed KRAS-FMe ($K_D = 98$ and 128 nM, respectively), whereas CRD showed no binding to non-processed KRAS as well as fully processed KRAS-FMe (**Supplementary Fig. 6a, b**). In another SPR experiment, we captured CRD and RBDCRD on the chip surface and flowed active KRAS-FMe or peptide containing farnesylated hypervariable region (HVR-FMe) over the surfaces. KRAS-FMe bound to RBDCRD ($K_D = 84$ nM) but not to CRD, and the HVR-FMe peptide showed no binding to either CRD or RBDCRD (**Supplementary Fig. 6c, d**). Cumulatively, these experiments indicate that when CRD is present as an isolated domain, we do not detect a binding interaction with either the farnesyl group or G-domain of KRAS under the conditions used in our experiments. However, when CRD is a part of RBDCRD, it increases the binding affinity to KRAS through the formation of multiple hydrogen bonds and non-bonded interactions (**Fig. 1b** and **Supplementary Table 5**).

Supplementary Figure 6: SPR binding analysis to examine the role of the farnesylated hypervariable region of KRAS in KRAS-RAF1(CRD) interaction. (a, b) Steady-state binding isotherms derived from the SPR data of RAF1(CRD) and RAF1(RBDCRD) binding to (a) non-processed KRAS G-domain and (b) fully processed KRAS (KRAS-FMe). **(c, d)** Steady-state binding isotherms derived from the SPR data of KRAS-FMe and HVR-FMe peptide binding to (c) RAF1(CRD) and (d) RAF1(RBDCRD).

Also, the use of RAF1(RBDCRD) polypeptide in SPR experiments makes it difficult to observe the effects of mutations of KRAS and CRD on the binding affinity because the CRD binding is masked by the predominant binding at RBD (actually modest effects are observed). Use of isolated CRD polypeptide would be recommended for this purpose.

Reply: We agree with the reviewer's suggestions. However, the binding affinity measurement using SPR between KRAS and isolated RAF1-CRD showed no measurable interaction. This is aligned with the previous report, where interaction between KRAS and isolated CRD has been suggested to be in the low micromolar range¹⁴. Therefore, it was not possible to see the effect of mutations of KRAS and CRD using isolated CRD. Because of this, we decided to use RAF1(RBDCRD) instead of isolated RAF1(CRD) in our SPR assay. Despite the RBD playing a predominant role in KRAS-RAF1 interaction, we see the effect (although modest) of KRAS and CRD mutations on KRAS-RAF1(RBDCRD) interaction.

Moreover, it is puzzling that the sensorgrams of KRAS-RAF1 binding presented in Supplementary Figures 1, 2 and 4 are all square shaped, which appears to have forced the authors to determine the K_D values by an unusual method measuring the dose-dependent increase of the binding

signal, not by a standard method measuring binding kinetics (k_a and k_d), in a Biacore machine. The authors must explain the reason why the sensorgrams are all square shaped and why they don't use the standard method of SPR measurement.

Reply: The reviewer raises an important point regarding the appropriate methods to use when analyzing Biacore data. One of the advantages of biosensor data is that it provides kinetic information of the binding interactions when mass transport limitations are minimized. Under rapid binding kinetics, Biacore sensorgrams appear as square-shaped profiles indicating the very fast k_a and k_d rates, as observed for RBD and some RBDCRD interactions in this study. Indeed, this is not unexpected as previous work using stopped-flow spectroscopy demonstrated rapid binding kinetics between RBD and RAS with k_a rates of $\sim 4 \times 10^7 \text{ M}^{-1} \text{ s}^{-1}$ and k_d rates of $\sim 10 \text{ s}^{-1}$ (Sydor *et al.*, *Biochemistry*, 1998, 37:14292-14299).

With interactions that have rapid kinetics, the two interacting partners rapidly reach a steady-state binding equilibrium, as exemplified by the square-shaped sensorgrams. Under these conditions, the equilibrium dissociation constant (K_D) can be calculated by plotting the total binding response as a function of the injected analyte (as per the Biacore application handbook section 2.2.2, <https://www.cytivalifesciences.co.jp/contact/pdf/BiacoreAssayHandbook.pdf>). Preliminary kinetic analysis of our RBD/RBDCRD binding data indicated that the off-rates were close to the detection limit of the instrument. In addition, while we used lower surface densities (≥ 150 RU of ligand) and flow rates of 30 ml/min, we did not evaluate whether the kinetics were impacted by transport limitations under these conditions. Finally, we found that several RBDCRD mutants have slower off-rates that were within the detection limit but could not be fit with a single off-rate. Rather than investigate the nature of the complex dissociation behavior, which we did not believe was biologically significant, we chose to use a steady-state analysis to calculate the K_D values.

REVIEWERS' COMMENTS

Reviewer #1 (Remarks to the Author):

The revised manuscript is significantly improved from the first submission and the authors have done a great job to address questions and concerns from all the reviewers. In addition to results from physical experiments to further characterize the impact of KRas-CRD interaction on RAF1 activation, the authors have also presented their work of modeling the Ras-RBD-CRD:membrane interaction in the context of existing literature and molecular mechanisms of MAPK activation, especially in the context of the still intriguing idea of Ras-dimerization.

This manuscript is a significant advance in the field of Ras induced RAF activation and implications for ERK signaling. As such, I recommend rapid publication of the article in its current form.

A few minor points:

(1) Figure 1b - are the error bars too small to see or are the curves a representative of repeated experiments? Please mention clarification in the figure legend.

(5) Figure 5c vs 5a: There appears to be a gel mobility shift difference among the KRAS Q61L in some of the experiments in 5c, whereas in figure 5a, for KRas WT, KRas in all experiments runs similarly on the gel. Is there a gel artifact in KRas Q61L context or is it a PTM that is yet to be characterized? This may not be crucial for this manuscript, but that would be interesting to address if that gel shift is not an artifact.

Reviewer #3 (Remarks to the Author):

This reviewer is satisfied with the authors' responses to the raised suggestions and questions especially those regarding the farnesylation requirements for Ras-CRD interaction and the SPR measurements. The quality of this revised manuscript is very much improved and suitable for publication in Nature Communications.

REPLY TO REVIEWERS' COMMENTS

Reviewer #1 (Remarks to the Author):

The revised manuscript is significantly improved from the first submission and the authors have done a great job to address questions and concerns from all the reviewers. In addition to results from physical experiments to further characterize the impact of KRas-CRD interaction on RAF1 activation, the authors have also presented their work of modeling the Ras-RBD-CRD:membrane interaction in the context of existing literature and molecular mechanisms of MAPK activation, especially in the context of the still intriguing idea of Ras-dimerization.

This manuscript is a significant advance in the field of Ras induced RAF activation and implications for ERK signaling. As such, I recommend rapid publication of the article in its current form.

Reply: We thank the reviewer for finding our revised manuscript a significant advance in the field and recommending its rapid publication.

A few minor points:

(1) Figure 1b - are the error bars too small to see or are the curves a representative of repeated experiments? Please mention clarification in the figure legend.

Reply: The curves in Figure 1b are representative of the SPR experiment, and we have now clarified this in the figure legend.

(5) Figure 5c vs 5a: There appears to be a gel mobility shift difference among the KRAS Q61L in some of the experiments in 5c, whereas in figure 5a, for KRas WT, KRas in all experiments runs similarly on the gel. Is there a gel artifact in KRas Q61L context or it is a PTM that is yet to be characterized? This may not be crucial for this manuscript, there that would be interesting to address if that gel shift is not an artifact.

Reply: Experiments described in Figures 5a and 5c had different KRAS mutants. For results shown in Figure 5a, we used active KRAS Q61L mutant for examining activation in RAF1 (RBD-CRD) mutants. In contrast, for the results shown in Figure 5c, we used six different KRAS mutants in the KRAS Q61L background to examine their effect on RAF activation. It does look like two KRAS mutants (K42A and V45E) shown in Figure 5c show gel mobility shift difference compared to other KRAS mutants. In the literature, we have not seen any report of a PTM at these two sites. It is likely to be a gel artifact, and it is not uncommon to see a small gel mobility shift difference among different mutants.

Reviewer #3 (Remarks to the Author):

This reviewer is satisfied with the authors' responses to the raised suggestions and questions especially those regarding the farnesylation requirements for Ras-CRD interaction and the SPR measurements. The quality of this revised manuscript is very much improved and suitable for publication in Nature Communications.

Reply: We thank the reviewer for finding our revised manuscript much improved and suitable for publication in Nature Communications.